# Exploring the cloud of feature interaction scores in a Rashomon set

**Sichao Li, Rong Wang, Quanling Deng & Amanda S. Barnard**
School of Computing
Australian National University
`{sichao.li,rong.wang,quanling.deng,amanda.s.barnard}@anu.edu.au`

## Abstract

Interactions among features are central to understanding the behavior of machine learning models. Recent research has made significant strides in detecting and quantifying feature interactions in single predictive models. However, we argue that the feature interactions extracted from a single pre-specified model may not be trustworthy since: *a well-trained predictive model may not preserve the true feature interactions and there exist multiple well-performing predictive models that differ in feature interaction strengths*. Thus, we recommend exploring feature interaction strengths in a model class of approximately equally accurate predictive models. In this work, we introduce the feature interaction score (FIS) in the context of a Rashomon set, representing a collection of models that achieve similar accuracy on a given task. We propose a general and practical algorithm to calculate the FIS in the model class. We demonstrate the properties of the FIS via synthetic data and draw connections to other areas of statistics. Additionally, we introduce a `Halo` plot for visualizing the feature interaction variance in high-dimensional space and a `swarm` plot for analyzing FIS in a Rashomon set. Experiments with recidivism prediction and image classification illustrate how feature interactions can vary dramatically in importance for similarly accurate predictive models. Our results suggest that the proposed FIS can provide valuable insights into the nature of feature interactions in machine learning models.

## 1 Motivation and prior work

Understanding the behavior of machine learning models is receiving considerable attention. Many researchers seek to identify what features are important and to describe their effects. Recently, several works have taken one step further from explaining feature attribution towards interaction attribution. This shift aims to explore the interplay between features and uncover how their interactions shape the behavior of the model. Such interaction occurs when the influence of one feature, denoted as $\boldsymbol{x}_i$, on the model's prediction can not be decomposed into a sum of subfunctions that do not involve the corresponding feature Sorokina et al. (2008); Friedman & Popescu (2008); Lou et al. (2013); Tsang et al. (2021), denoted as $\boldsymbol{X}_{\setminus i}$, statistically defined as: $f \neq \sum_{i \in \mathcal{I}} f_i(\boldsymbol{X}_{\setminus i})$. For example, there exists feature interaction effects in $sin(\cdot)$ function between $\boldsymbol{x}_1$ and $\boldsymbol{x}_2$, where $sin(\boldsymbol{x}_1 + \boldsymbol{x}_2) \neq f_1(\boldsymbol{x}_2) + f_2(\boldsymbol{x}_1)$. Higher-order interactions among features are defined similarly. The feature interaction can provide a more comprehensive and accurate understanding of the underlying factors driving model predictions, leading to robust and improved models; prominent applications include recommendation systems Guo et al. (2017); Tsang et al. (2020a), DNA sequence analysis and modeling Greenside et al. (2018), and text-image information retrieval Wang et al. (2019).

Existing approaches that detect and explain the feature interactions mainly include (a) traditional statistical approaches such as analysis of variance (ANOVA) and H-statistics Fisher (1992); Mandel (1961); Friedman & Popescu (2008); Greenwell et al. (2018) and (b) more recent machine learning model-based methods such as Additive Groves (AG) Sorokina et al. (2008). AG is a non-parametric method of identifying interactions by imposing structural constraints on an additive model of regression trees that compare the performance of unrestricted and restricted prediction models. LASSO-based methods Bien et al. (2013) select interactions by shrinking the coefficients of insignificant terms to zero. Specifically, neural network-based models, e.g. feed-forward neural

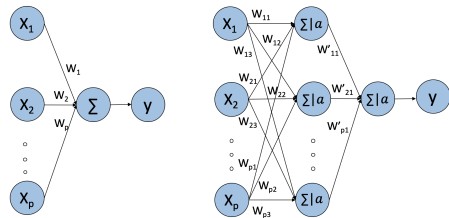
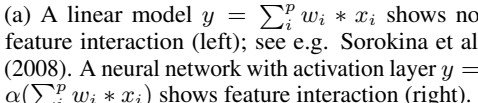
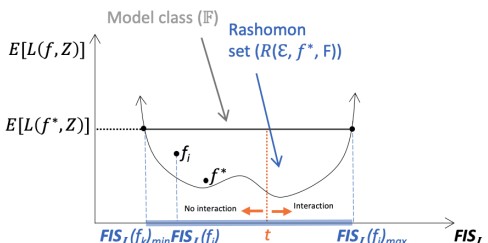

(a) A linear model $y = \sum_i^p w_i * x_i$ shows no feature interaction (left); see e.g. Sorokina et al. (2008). A neural network with activation layer $y = \alpha(\sum_i^p w_i * x_i)$ shows feature interaction (right).

(b) The $x$-axis shows the FIS for the feature set $\mathcal{I}$ in each model $f$ while the $y$-axis shows the expected loss of the model in the model class. The FISC range is highlighted in blue.

Figure 1: Two accurate models with different feature interactions and an illustration of FIS in a hypothetical Rashomon set $\mathcal{R}(\epsilon, f^*, \mathcal{F})$.

networks, Bayesian neural networks and convolutional neural networks, are utilized to examine the weight matrices Tsang et al. (2021); Singh et al. (2019); Cui et al. (2020). Several more advanced explainable frameworks have been proposed to capture feature interactions from well-defined models, e.g. Shapley Taylor Interaction Index (STI) Grabisch & Roubens (1999); Sundararajan et al. (2020); Integrated Hessians (IH) Janizek et al. (2021) and `Archipelago` Tsang et al. (2020b).

**Limitations of previous work**  Previous works can be summarized as detecting and explaining feature interaction effects in a single trained model $f$. Here, we argue that *the high accuracy of a well-trained model can not guarantee the preservation of true feature interactions.* As a simple example, we consider the two error-free models $f_1(a, b, c) = \frac{-b + \sqrt{b^2 - 4ac}}{2a}$ and $f_2(a, b, c) = \frac{-b - \sqrt{b^2 - 4ac}}{2a}$ for finding a root of the quadratic equation $ax^2 + bx + c = 0$; the input variables $a$, $b$, and $c$ exhibit different feature interactions in these two models as there is a sign difference in the numerator. See Fig. 1(a) for another general example where a well-performing linear model with no feature interaction might be modeled using a deep neural network with the same model performance but with feature interaction. A detailed proof of concept is provided in Appendix C.

In general, for a given task, there exists a class of equally accurate models that lead to different feature interactions. To address the identified limitations, one potential solution is to analyze the feature interaction in a set of well-performing models. Emerging research Rudin (2019); Fisher et al. (2019); Dong & Rudin (2020); Li & Barnard (2022a) has demonstrated the necessity of examining feature importance not just within a single well-trained model but across a set of high-performing models, but none of these works delved into a comprehensive analysis of feature interactions [1].

**Our contributions**  In this paper, we study feature interaction and demonstrate that investigating feature interactions in a single model may fall short of accurately identifying the complex relationships between features. A more meaningful approach involves exploring feature interactions within a class of models. Our primary contributions can be summarized as:

- We introduce feature interaction score (FIS) as a means to quantify the feature interaction strength in a pre-specified predictive model. In Sec. 2.1 and Sec. 4.1, we establish and validate the connection between FIS and the existing literature.

- We consider FIS in a model set and introduce FIS cloud (FISC) in Sec. 2.1. We demonstrate in Sec. 4 how feature interactions contribute to predictions and how feature interactions can vary across models with reasonable and explainable accurate predictions in terms of FIS.

- We present a non-linear example in multilayer perceptron (MLP) to mathematically characterize the Rashomon set and FISC. For the general case, we propose a greedy search algorithm to explore FIS and to characterize FISC in a Rashomon set with two novel visualization tools, namely `Halo` and `swarm` plots.

---

[1] More related works are discussed in the Appendix A

## 2 FEATURE INTERACTION SCORE AND THE CLOUD OF SCORES

The terms "statistical interaction", "feature interaction", and "variable interaction" in the literature are highly related (Tsang et al., 2021; Greenwell et al., 2018; Sorokina et al., 2008; Tsang et al., 2020b). Here, we use the term "feature interaction" and "feature interaction score" to indicate a statistical non-additive interaction of features in a model. In general, we use bold lowercase letters such as $\boldsymbol{v}$ to represent a vector and $v_i$ denotes its $i$-th element. Let the bold uppercase letters such as $\boldsymbol{A}$ denote a matrix with $a_{[i,j]}$ being its $i$-th row and $j$-th column entry. The vectors $\boldsymbol{a}_{[i,\cdot]}$ and $\boldsymbol{a}_{[\cdot,j]}$ are its $i$-th row and $j$-th column, respectively. Let $(\boldsymbol{X}, \boldsymbol{y}) \in \mathbb{R}^{n \times (p+1)}$ denote the dataset where $\boldsymbol{X} = [\boldsymbol{x}_{[\cdot,1]}, \boldsymbol{x}_{[\cdot,2]}, ..., \boldsymbol{x}_{[\cdot,p]}]$ is a $n \times p$ covariate input matrix and $\boldsymbol{y}$ is a $n$-length output vector. Let $\mathcal{I}$ be a subset of feature indices: $\mathcal{I} \subset \{1, 2, ..., p\}$ and its cardinality is denoted by $|\mathcal{I}|$. All possible subsets are referred as $\mathbb{I} = \{\mathcal{I} \mid \mathcal{I} \subset \{1, 2, ..., p\}\}$. In the context of no ambiguity, we drop the square brackets on the vector and simply use $\boldsymbol{x}_s$ to denote the feature. $\boldsymbol{X}_{\backslash s}$ is the input matrix when the feature of interest (denoted as $s$ here) is replaced by an independent variable. Let $f : \mathbb{R}^{n \times p} \to \mathbb{R}^n$ be a predictive model and $L : (f(\boldsymbol{X}), \boldsymbol{y}) \to \mathbb{R}$ be the loss function. The expected loss and empirical loss are defined as $L_{exp} = \mathbb{E}[L(f(\boldsymbol{X}), \boldsymbol{y})]$ and $L_{emp} = \sum_{i=1}^{n} L(f(\boldsymbol{x}_{[i,\cdot]}), y_i)$, respectively.

### 2.1 DEFINITIONS

**Mask-based Rashomon set**   Our goal is to explore feature interactions in a set of well-performing models, also known as a Rashomon set (Fisher et al., 2019). For a given pre-specified class of predictive models $\mathcal{F} \subset \{f \mid f : \boldsymbol{X} \to \boldsymbol{y}\}$, the Rashomon set is defined as

$$\mathcal{R}(\epsilon, f^*, \mathcal{F}) = \{f \in \mathcal{F} \mid \mathbb{E}[L(f(\boldsymbol{X}), \boldsymbol{y})] \leq \mathbb{E}[L(f^*(\boldsymbol{X}), \boldsymbol{y})] + \epsilon\}, \tag{1}$$

where $f^*$ is a trained model (or reference model), and $\epsilon > 0$ is a pre-defined acceptable loss tolerance. Given the existence of equivalent simple and complex models discussed in Semenova et al. (2022), one way to characterize a model is to concatenate a layer, denoted as $m \in \mathbb{R}$, into a backbone model. That is,

$$\forall f \in \mathcal{F} \text{ with } \mathbb{E}[L(f(\boldsymbol{X}), \boldsymbol{y})], \quad \exists m \text{ s.t. } \mathbb{E}[L(f \circ m(\boldsymbol{X}), \boldsymbol{y})] \leq \mathbb{E}[L(f^*(\boldsymbol{X}), \boldsymbol{y})] + \epsilon. \tag{2}$$

With this "input pre-processing" layer, we refer to the resulting Rashonmon set as the *Mask-based Rashomon set* and claim that there always exists an alternative for any model in the Rashomon set, logically expressed as: $\forall f \in \mathcal{F} (\mathbb{E}[L(f(\boldsymbol{X}), \boldsymbol{y})] \to (\exists m (\mathbb{E}[L(f^* \circ m(\boldsymbol{X}), \boldsymbol{y})] \simeq \mathbb{E}[L(f(\boldsymbol{X}), \boldsymbol{y})])$. See Appendix B for the detailed natural deduction.

**Feature interaction score (FIS)**   The common way to quantify feature interaction based on the statistical non-additivity Lou et al. (2013) is to measure the performance change of feature importance from the baseline (Sundararajan et al., 2017; Janizek et al., 2021). The performance change can be decomposed into main feature effects and feature interaction effects. Several feature attribution methods have identified the necessity of a baseline value representing a lack of information when generating explanations. We refer to individual feature effects which do not interact with other features as the main effect. Inspired by model reliance and similar methods Fisher et al. (2019); Zhu et al. (2015); Gregorutti et al. (2017) that measure the change in the loss by replacing the variable of interest with a new random independent variable, we denote by $\varphi_i(f) = \mathbb{E}[L(f(\boldsymbol{X}_{\backslash i}), \boldsymbol{y})] - \mathbb{E}[L(f(\boldsymbol{X}), \boldsymbol{y})]$ and $\varphi_{\mathcal{I}}(f) = \mathbb{E}[L(f(\boldsymbol{X}_{\backslash \mathcal{I}}), \boldsymbol{y})] - \mathbb{E}[L(f(\boldsymbol{X}), \boldsymbol{y})]$ as the main effect and joint effect respectively. Herein, $\mathbb{E}[L(f(\boldsymbol{X}), \boldsymbol{y})]$ is the baseline effect that provides interpretability. In practice, we usually permute the features of interest multiple times to achieve a similar measurement (Datta et al., 2016).

With this in mind, the FIS is defined as the difference between the loss change of replacing multiple features simultaneously and the sum of the loss change of replacing multiple features individually:

$$FIS_{\mathcal{I}}(f) = \varphi_{\mathcal{I}}(f) - \sum_{i \in \mathcal{I}} \varphi_i(f). \tag{3}$$

**Feature interaction score cloud (FISC)**   We can generate the set of FISs as: $FIS_{\mathbb{I}}(f) = \{FIS_{\mathcal{I}_1}(f), FIS_{\mathcal{I}_2}(f), ..., FIS_{\mathcal{I}_{2^p}}(f)\}$. Thus, the FIS of certain feature set $\mathcal{I}$ in the Rashomon set $\mathcal{R}(\epsilon, f^*, \mathcal{F})$ is given by $FISC_{\mathcal{I}}(\mathcal{R}) = \{FIS_{\mathcal{I}}(f) : f \in \mathcal{R}\}$, illustrated in Fig. 1 (b). We refer to the set of FISs with respect to a feature set in a model class as FISC. This cloud of scores has the range

$$[FISC_{\mathcal{I}}(\mathcal{R})_{min}, FISC_{\mathcal{I}}(\mathcal{R})_{max}] = [\min_{f \in \mathcal{R}} FIS_{\mathcal{I}}(f), \max_{f \in \mathcal{R}} FIS_{\mathcal{I}}(f)]. \tag{4}$$

The complete set of FISs in a Rashomon set is $FISC_{\mathbb{I}}(\mathcal{R}) = \{FIS_{\mathbb{I}}(f) : f \in \mathcal{R}\}$.

## 2.2 RELATION TO EXISTING FEATURE INTERACTION ATTRIBUTION APPROACH

Since FISC focuses on explaining feature interactions across a set of models rather than a single model, existing explainable methods for a single model should be viewed as a special case of our framework. We demonstrate that the state-of-the-art (SOTA) `ArchDetect` method Tsang et al. (2020b) can be derived from our approach; see Appendix D.

## 3 COMPUTATIONAL CHALLENGES OF FISC

The first challenge is establishing the theoretical upper and lower bound of FISC for non-linear models characterized by uncertain, potentially non-convex loss functions. The second challenge arises from the NP-hard nature of searching the Rashomon set defined by hypothesis space Semenova et al. (2022); Hsu & Calmon (2022). We first show that it is possible to find the boundary of FISC under certain assumptions and conditions in Sec. 3.1, aligning with previous work in this area (Fisher et al., 2019; Dong & Rudin, 2020). Then we present a sampling algorithm for more general scenarios in Sec. 3.2, where model structures and loss functions might be uncertain. This approach enables us to empirically approximate the FISC.

### 3.1 AN EXAMPLE APPLICATION OF FISC IN MULTILAYER PERCEPTRON (MLP)

Here, we consider a non-trivial MLP with a sigmoid activation function as an example. Let $f : \boldsymbol{X} \to \boldsymbol{y}$ be a predictive model defined as

$$f(\boldsymbol{X}) = \boldsymbol{\alpha}^T \frac{1}{1 + e^{-\boldsymbol{\beta}^T \boldsymbol{X}}} + b. \tag{5}$$

The expected root mean squared error (RMSE) loss is $\mathbb{E}[L(f(\boldsymbol{X}), \boldsymbol{y})] = \mathbb{E}[\sqrt{(\boldsymbol{y} - f(\boldsymbol{X}))^2}]$ that is mathematically equivalent to $\mathbb{E}[\boldsymbol{y} - f(\boldsymbol{X})]$ or $\mathbb{E}[f(\boldsymbol{X}) - \boldsymbol{y}]$. For simplicity, we consider the case $\mathbb{E}[\boldsymbol{y} - f(\boldsymbol{X})]$ to derive the condition on the mask $\boldsymbol{m}$ to characterize the Rashomon set. Without loss of generality, we assume $\boldsymbol{\alpha} > 0$ and a scaling on $\epsilon$ by $\boldsymbol{\alpha}$. We further introduce an assumption[2] that $\epsilon \leq \min\{\mathbb{E}[\frac{1}{1 + e^{-\boldsymbol{m}^T \cdot \boldsymbol{\beta}^T \boldsymbol{X}}}], \mathbb{E}[\frac{e^{-\boldsymbol{m}^T \cdot \boldsymbol{\beta}^T \boldsymbol{X}}}{1 + e^{-\boldsymbol{m}^T \cdot \boldsymbol{\beta}^T \boldsymbol{X}}}]\}$. Invoking Eq. 2, this assumption allows us to derive the following inequality

$$\mathbb{E}\left[ \ln \left( \frac{1 - \epsilon - \epsilon e^{-\boldsymbol{\beta}^T \boldsymbol{X}}}{e^{-\boldsymbol{\beta}^T \boldsymbol{X}} + \epsilon + \epsilon e^{-\boldsymbol{\beta}^T \boldsymbol{X}}} \right) \right] \leq \mathbb{E}[(\boldsymbol{\beta}^T \boldsymbol{X})\boldsymbol{m}^T] \leq \mathbb{E}\left[ \ln \left( \frac{1 + \epsilon + \epsilon e^{-\boldsymbol{\beta}^T \boldsymbol{X}}}{e^{-\boldsymbol{\beta}^T \boldsymbol{X}} - \epsilon - \epsilon e^{-\boldsymbol{\beta}^T \boldsymbol{X}}} \right) \right], \tag{6}$$

which gives the condition on $\boldsymbol{m}$ that characterises the mask-based Rashomon set. We refer to Appendix E for its derivation.

### 3.2 GREEDY SEARCH FOR EMPIRICAL FISC LOWER AND UPPER BOUNDS

Unfortunately, due to the nonlinear nature of FIS, it is difficult to get a closed-form expression of the exact hypothetical bounds in general, especially when $f$ and $L$ functions vary. Here we propose a general way to sample from the Rashomon set and approximate the range of FISs. The RMSE loss function and a pairwise feature interaction $\mathcal{I} = \{i, j\}$ are chosen to illustrate the idea, as RMSE will show the connection between our method and `ArchDetect` in `Archipelago`.

#### 3.2.1 SAMPLING FROM A MASK-BASED RASHOMON SET

Instead of searching for all possible models, we sample models from the mask-based Rashomon set by concatenating a multi-valued mask $\boldsymbol{m} = \{m_1, m_2, \cdots, m_p\}$ into a backbone $f$, resulting in a new model $\mathcal{M}_{\boldsymbol{m}} = f \circ \boldsymbol{m}$. We can sample the Rashomon set by exploring different $\boldsymbol{m}$ such that

$$\mathcal{R}(\epsilon, f, \mathcal{F}) = \{\mathcal{M}_{\boldsymbol{m}} \in \mathcal{F} \mid \mathbb{E}[L(\mathcal{M}_{\boldsymbol{m}}(\boldsymbol{X}), \boldsymbol{y})] \leq \mathbb{E}[L(f(\boldsymbol{X}), \boldsymbol{y})] + \epsilon\}. \tag{7}$$

All models are characterized by $\boldsymbol{m}$ and only masks that potentially affect the FIS are of interest. Redundant models such as the composition of identity functions can be ignored to save resources.

---

[2] A smaller $\epsilon$ means a smaller Rashomon set with models closer to the reference well-trained model $f^*$.

We illustrate this through a one-layer mask, denoted by $\boldsymbol{m}_{\mathcal{I}}$ with the component of feature $\boldsymbol{x}_i$:

$$(\boldsymbol{m}_{\mathcal{I}})_i = \begin{cases} m_i & i \in \mathcal{I}, \\ 1 & i \notin \mathcal{I}. \end{cases} \tag{8}$$

The FIS defined in Eq. 3 with the above setting can be rewritten as:

$$FIS_{\mathcal{I}}(\mathcal{M}_{\boldsymbol{m}_{\mathcal{I}}}) = \varphi_{\mathcal{I}}(\mathcal{M}_{\boldsymbol{m}_{\mathcal{I}}}) - \sum_{i \in \mathcal{I}} \varphi_i(\mathcal{M}_{\boldsymbol{m}_{\mathcal{I}}}). \tag{9}$$

In practice, instead of calculating all possible $\mathbb{I}$, we can calculate all main effects in the feature space $\{\mathcal{M}_{\boldsymbol{m}_i} \in \mathcal{F} \mid \varphi_i(\mathcal{M}_{\boldsymbol{m}_i})\}_{i=1}^p$ first and then inversely calculate any order of interaction by decomposing $\boldsymbol{m}_{\mathcal{I}} = \prod_{i \in \mathcal{I}} \boldsymbol{m}_i$.

### 3.2.2 GREEDY SEARCH ALGORITHM

To search all possible $(\boldsymbol{m}_i)_{i=1}^p$ for features' main effects within the Rashomon set, we propose a greedy search algorithm. For each feature's main effect $\varphi_i(\mathcal{M}_{\boldsymbol{m}_i})$, we set two $p$-length vectors of ones $\boldsymbol{m}_{i+}$ and $\boldsymbol{m}_{i-}$ for upper bound search and for lower bound search, as both actions can increase loss values if the model $f$ is well-trained. During the search process, we set a uniform learning rate to regulate the number of models generated for all dimensions. Higher learning rates lead to a reduced number of models. The searching process continues until the loss condition is not satisfied $\mathcal{M}_{\boldsymbol{m}_i} \notin \mathcal{F}$. This condition is imposed on the loss difference, denoted as $\phi_i = \mathbb{E}[L(\mathcal{M}_{\boldsymbol{m}_i}(\boldsymbol{X}), \boldsymbol{y})] - \mathbb{E}[L(f(\boldsymbol{X}), \boldsymbol{y})]$. Here, each mask corresponds to a concatenated model, resulting in an associated loss value. Any mask $\boldsymbol{m}_i$ during training meets $\mathcal{M}_{\boldsymbol{m}_i} \in \mathcal{F}$ and is one of the target models for feature $\boldsymbol{x}_i$ in $\mathcal{R}(\epsilon, f^*, \mathcal{F})$. Finally, we can obtain a model class with main effects $\{\mathcal{M}_{\boldsymbol{m}_i} \in \mathcal{F} \mid \varphi_i(\mathcal{M}_{\boldsymbol{m}_i})\}_{i=1}^p$, and calculate any order of feature interaction scores: $FIS_{\mathcal{I}}(\mathcal{R}) = \{FIS_{\mathcal{I}}(\mathcal{M}_{\boldsymbol{m}_{\mathcal{I}}}) : \mathcal{M}_{\boldsymbol{m}_{\mathcal{I}}} \in \mathcal{R}\}$ with Eq. 9; see the pseudocode in Algorithm 1.

### 3.2.3 THE INFLUENCE OF JOINT EFFECTS ON THE BOUNDARY OF THE RASHOMON SET

When searching for all main effects of features in the Rashomon set through the loss difference, we isolate the feature of interest and record the loss difference to ensure the Rashomon set condition. In theory, the joint effects of features should be the sum of main effects of features when there is no feature interaction. If the sum of the main effects is less than $\epsilon$, then the joint effect would also be less than $\epsilon$. However, in reality, the presence of feature interaction causes the joint effects to surpass the boundary. This motivates us to study the influences of interaction and the extent to which it varies.

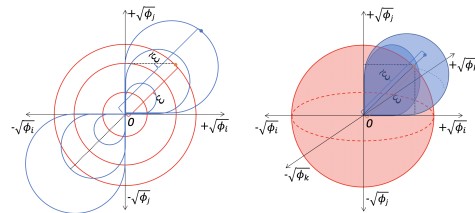

Figure 2: Exploring feature interaction of function $f = x_i + x_j + x_i * x_j$ and $f = x_i + x_j + x_k + x_i * x_j * x_k$ in Rashomon set. The data points are sampled from both functions. Exploring $\phi_i$ and $\phi_j$ separately enables us to draw red circles representing $\sum_{i \in I} \phi_i = \epsilon$, where $\epsilon$ is the radius, e.g. $\epsilon = 0.1$ can be potentially expressed by $\phi_i = 0.01$ and $\phi_j = 0.09$; a detailed example is provided in Appendix G. From inner to outer, the radii are 0.1, 0.2, and 0.3, respectively and the corresponding joint effects $\tilde{\epsilon}$ can be calculated by $\phi_{i,j}$ and $\phi_{i,j,k}$, from which we plot the blue curves. This can be achieved in 2D (left) and higher dimensions, e.g. 3D (right).

**Algorithm 1** Greedy search algorithm

---

**Require:** learning rate $> 0 \wedge \epsilon \geq 0$
**Ensure:** $L(f \circ \boldsymbol{m}(\boldsymbol{X}), \boldsymbol{y}) \leq L(f^*(\boldsymbol{X}), \boldsymbol{y}) + \epsilon$
    $\boldsymbol{m}_+ = \mathbb{1}_{i=1}^p, \boldsymbol{m}_- = \mathbb{1}_{i=1}^p, \text{counter} = 0$
    $\text{loss}_{ref} \Leftarrow L(f^*(\boldsymbol{X}), \boldsymbol{y})$
    **while** counter $\leq 4$ **do**
        **if** searching for the upper bound of $m$ **then**
            $\boldsymbol{m}'_+ \Leftarrow \text{Copy}(\boldsymbol{m}_+)$
            $\boldsymbol{m}'_+[s] \Leftarrow \boldsymbol{m}'_+[s] + \text{learning rate}$
            $\phi_s \Leftarrow L(f^* \circ \boldsymbol{m}'_+(\boldsymbol{X}_s), y) - \text{loss}_{ref}$
            **if** $\phi_s \leq \epsilon$ **then**
                $\boldsymbol{m}_+ \Leftarrow \boldsymbol{m}'_+$
            **else**
                learning rate $\Leftarrow$ learning rate $\times 0.1$
                counter $\Leftarrow$ counter $+ 1$
        **else** searching for the lower bound of $m$
            $\boldsymbol{m}'_- \Leftarrow \text{Copy}(\boldsymbol{m}_-)$
            $\boldsymbol{m}'_-[s] \Leftarrow \boldsymbol{m}'_-[s] - \text{learning rate}$
            $\phi_s \Leftarrow L(f^* \circ \boldsymbol{m}'_-(\boldsymbol{X}_s), y) - \text{loss}_{ref}$
            **if** $\phi_s \leq \epsilon$ **then**
                $\boldsymbol{m}_- \Leftarrow \boldsymbol{m}'_-$
            **else**
                learning rate $\Leftarrow$ learning rate $\times 0.1$
                counter $\Leftarrow$ counter $+ 1$

---

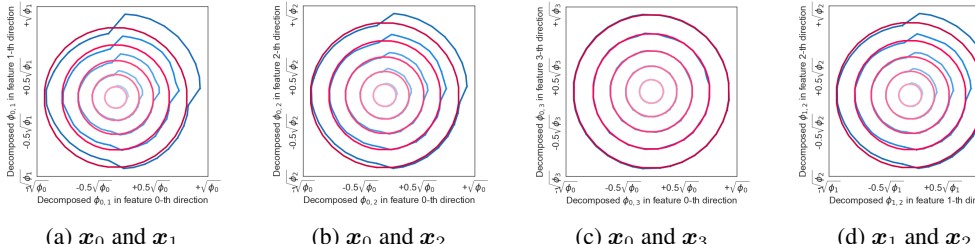

(a) $\boldsymbol{x}_0$ and $\boldsymbol{x}_1$    (b) $\boldsymbol{x}_0$ and $\boldsymbol{x}_2$    (c) $\boldsymbol{x}_0$ and $\boldsymbol{x}_3$    (d) $\boldsymbol{x}_1$ and $\boldsymbol{x}_2$

Figure 3: Pairwise `Halo` plot from FISC applied to an MLP trained from NID setting. The ground truth interaction defined in function $f(x) = \bigwedge(x; \{x_0^*, x_1^*\} \cup x_2') + \bigwedge(x; \{x_i^*\}_{i=11}^{30}) + \sum_{j=1}^{40} x_j$ from left to right are: true, true, false, true, respectively.

**`Halo` plot**    A `Halo` plot fixes the sum of main effects within the boundary of the Rashomon set and visualizes the change in the joint effect. The fixed loss change is formulated as: $\sum_{i \in \mathcal{I}} \phi_i(\mathcal{M}_{\boldsymbol{m}_i}, f) = t$, where $t \in (0, \epsilon]$ to ensure the loss condition. Then we map an $n$-sphere with radius $t$ and plot corresponding empirical loss $\phi_{i,j}(\mathcal{M}_{\boldsymbol{m}_{i,j}}, f) : \mathbb{E}[L(\mathcal{M}_{\boldsymbol{m}_{i,j}}(\boldsymbol{X}), \boldsymbol{y})] - \mathbb{E}[L(f(\boldsymbol{X}), \boldsymbol{y})]$, from which we can directly visualize how the joint effect varies and affects the Rashomon set boundary. Any order of interaction `Halo` plot can be similarly derived and two simple examples are used to illustrate the idea when $|\mathcal{I}| = 2$ (2D) and $|\mathcal{I}| = 3$ (3D) in Fig. 10. For higher order interaction visualizations, we discussed in Appendix F.

**Computational time**    Our framework effectively addresses the computational expense associated with calculating interactions, eliminating the need to calculate all pairs, especially in scenarios involving higher-order interactions. By performing a single-time search for all main effects and subsequently calculating the desired interactions, the proposed algorithm significantly reduces computational complexity. Specifically, a subset of FISC range $[FISC_{\mathcal{I}}(\mathcal{R})_{min}, FISC_{\mathcal{I}}(\mathcal{R})_{max}]$ bounded by $t$ can be approximated using representative points on $\sum_{i \in \mathcal{I}} \phi_i(\mathcal{M}_{\boldsymbol{m}_i}, f) = t$. Collecting ordered pairs $S = \{(\phi_i x, \phi_j y) \mid (x, y) \in \mathbb{R}, 0.1 \leq x, y \leq 0.9, x + y = 1\}$ is sufficient to characterize pairwise interactions, which requires $|S| \times 2^2$ calculations. Similarly, any order of interaction requires $|S| \times 2^{|\mathcal{I}|}$ model predictions.

## 4    EXPERIMENTS

In order to showcase the practical value of FISC, we have set two primary objectives and they are: (a) to demonstrate the concept of FISC using a well-trained model that overlooks true feature interactions, and (b) to examine the impact of feature interactions and explore the variation in FIS on accurate model predictions from our greedy sampling method. In the experiments below, we identify the main feature effects within the Rashomon set and calculate the FIS and FISC using representative models, followed by visualizations using `Halo` and `swarm` plots. Note that the meaning of axis in `Halo` plots refers to Sec. 3.2.3.

### 4.1    QUANTITATIVE COMPARISON AND EXTENSION

**Synthetic Validation**    In order to validate the performance of FIS, we use SOTA from Tsang et al. (2020b) to generate ground truth feature interactions and apply our framework to the synthetic functions in Table 1 in the same context. The interaction detection area under the curve (AUC) on these functions from different baseline methods is provided in Tables 1 and 2. As the relation to `ArchDetect` provided in Appendix D, our FIS is designed under certain conditions to detect the interactions of their proposed contexts and the results are as expected.

**Exploring FISC from an accurate MLP relying on "inaccurate" interactions**    The above accuracy is based on the knowledge of given functions, but in practice, we usually have a dataset with unknown functional relationships. This prompts us to consider how we account for the performance of a well-trained model that *achieves high accuracy in making predictions, yet fails to preserve ground truth interactions* with FISC. In this study, we conduct a general black box explanation task.

Specifically, we utilize a dataset generated by a model $F_4$ (assuming unknown to us) and train an MLP using the same neural interaction detection (NID) setting.

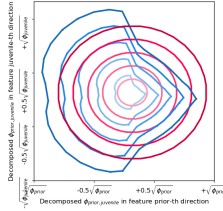

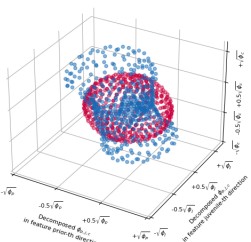

Figure 4: Pairwise `Halo`: 'prior' and 'juvenile' (top); and triplewise `Halo`: 'prior', 'juvenile' and 'charge' (bottom).

Table 1: Functions with ground truth interactions, where $\boldsymbol{x}^* = [1, 1, 1..., 1] \in \mathbb{R}^{40}$, $\boldsymbol{x}' = [-1, -1, -1..., -1] \in \mathbb{R}^{40}$ and $z[i] = x_i$ is a key-value pair function. $\bigwedge(x; z)$ is defined as 1 if index of $\boldsymbol{x}$ belongs to any keys of function $z$, otherwise $-1$.

$$F_1(x) = \sum_{i=1}^{10}\sum_{j=1}^{10} x_i x_j + \sum_{i=11}^{20}\sum_{j=21}^{30} x_i x_j + \sum_{k=1}^{40} x_k$$
$$F_2(x) = \bigwedge(x; \{x_i^*\}_{i=1}^{20}) + \bigwedge(x; \{x_i^*\}_{i=11}^{30}) + \sum_{j=1}^{40} x_j$$
$$F_3(x) = \bigwedge(x; \{x_i'\}_{i=1}^{20}) + \bigwedge(x; \{x_i^*\}_{i=11}^{30}) + \sum_{j=1}^{40} x_j$$
$$F_4(x) = \bigwedge(x; \{x_1^*, x_2^*\} \cup x_3') + \bigwedge(x; \{x_i^*\}_{i=11}^{30}) + \sum_{j=1}^{40} x_j$$

Table 2: Comparison of pairwise feature interaction detection AUC. FIS is set in the same context to detect feature interactions.

| Method | $F_1$ | $F_2$ | $F_3$ | $F_4$ |
|---|---|---|---|---|
| Two-way ANOVA | 1.0 | 0.51 | 0.51 | 0.55 |
| Neural Interaction Detection | 0.94 | 0.52 | 0.48 | 0.56 |
| Shapley Interaction Index | 1.0 | 0.50 | 0.50 | 0.51 |
| Shapley Taylor Interaction Index | 1.0 | 0.50 | 0.50 | 0.51 |
| ArchDetect | 1.0 | 1.0 | 1.0 | 1.0 |
| FIS in the context (this work) | **1.0** | **1.0** | **1.0** | **1.0** |

Table 3: MCR[3] comparison between results from VIC (right column under each feature) and results in bold from our greedy search algorithm (left column under each feature).

| MCR | Age | | Race | | Prior | | Gender | | Juvenile | | Charge | |
|---|---|---|---|---|---|---|---|---|---|---|---|---|
| AVG(MCR) | **1.113** | 1.021 | **0.969** | 1.013 | **1.330** | 1.048 | **0.951** | 1.009 | **1.367** | 1.038 | **1.065** | 1.004 |
| MAX(MCR+) | **1.141** | 1.060 | **1.007** | 1.037 | **1.372** | 1.116 | **0.986** | 1.032 | **1.390** | 1.097 | **1.095** | 1.021 |
| MIN(MCR-) | **1.072** | 0.987 | **0.919** | 0.994 | **1.123** | 0.988 | **0.900** | 0.989 | **1.278** | 0.992 | **1.004** | 0.990 |
| (MCR+)-(MCR-) | **0.068** | 0.073 | **0.088** | 0.043 | **0.249** | 0.129 | **0.086** | 0.043 | **0.112** | 0.105 | **0.091** | 0.031 |

Based on the ranking metrics AUC of the receiver operating characteristic curve (ROC), the MLP achieves an interaction detection accuracy of 56%. The trained MLP is set as the reference model and we explore FISC in the Rashomon set with $t = [0.2\epsilon, 0.4\epsilon, 0.6\epsilon, 0.8\epsilon, \epsilon]$ respectively, to observe that the range of each pairwise feature interaction exceeds the threshold extracted above; see Fig. 3 and Fig. 5(a). This provides insights into the role of FISC in bridging the gap between unknown ground truth function $f$ and a well-trained model that fails to preserve true feature interactions. The potential loss of feature interaction preservation in a model can be mitigated by other models in the Rashomon set. In this case, one can conclude that *there always exists a model in the Rashomon set in which certain features interact*.

## 4.2 RECIDIVISM PREDICTION

The use of the COMPAS score, as reported by the ProPublica news organization, has been instrumental in examining its applicability in proprietary recidivism prediction, as evidenced by previous studies (Flores et al., 2016; Rudin et al., 2020). However, it has been demonstrated that interpreting a pre-specified model has its limitations, as exemplified by the Rashomon set (Fisher et al., 2019; Dong & Rudin, 2020). While the Rashomon set has been studied, the significance of feature interactions within the set has not yet been explored.

### 4.2.1 COMPARISON WITH EXISTING RELEVANT LITERATURE

**Feature importance comparison-variable importance cloud (VIC)** Given the absence of an identical task in existing literature, we adopt the experiment settings utilized in prior research Dong &

---

[3]Model class reliance (MCR) is the range of feature importance values (main effect in our definition) in the Rashomon set, used by Dong & Rudin (2020) and Fisher et al. (2019)

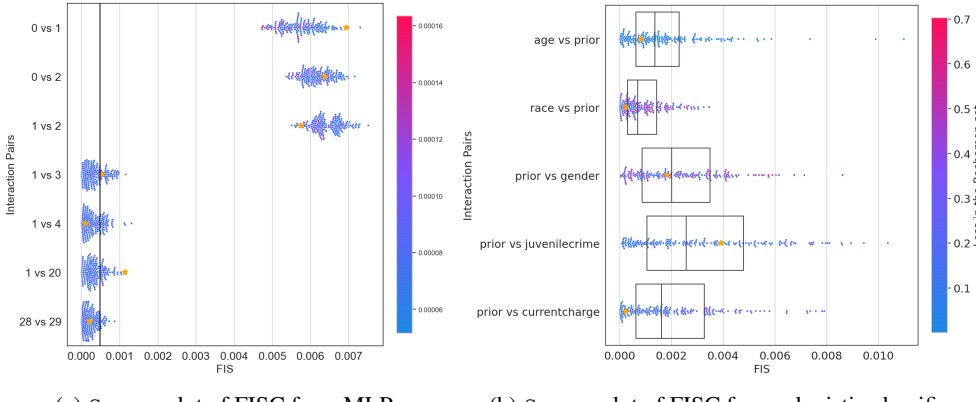

(a) `Swarm` plot of FISC from MLP        (b) `Swarm` plot of FISC from a logistic classifier

Figure 5: Illustration of FISC using `swarm` plots, where the black vertical line is the threshold and the yellow star is the reference FIS. Each point in the plot is coloured based on loss value.

Rudin (2020) that aligns with our research focus. We use the dataset of 7,214 defendants in Broward County, Florida Washington (2018), the same logistic model, and set $\epsilon = 0.05$. To guarantee a fair and unbiased comparison, we explore the Rashomon set using our greedy search algorithm and evaluate the feature importance using MCR. The resulting outcomes are presented in Table 3 and it demonstrates the promising range of feature importance in a general case. Although the feature importance is not the focus of our study, it's still worthy of further work. Dong Dong & Rudin (2020) proposed the information compensation structure and stated that information loss in "prior" was most efficiently compensated by 'juvenile', but the underlying reason was missing. We find that VIC can not capture the higher-order information compensation structure.

Table 4: Comparison of FISC calculated from different sampling methods (partial). We use *vs* to denote the interaction pairs and best results are highlighted in bold.

| Interaction pairs | AWP sampling | | Greedy sampling (ours) | | Random sampling | |
|---|---|---|---|---|---|---|
| | $FIS_{min}$ | $FIS_{max}$ | $FIS_{min}$ | $FIS_{max}$ | $FIS_{min}$ | $FIS_{max}$ |
| age *vs* race | -0.00026 | 0.000113 | **-0.00089** | **0.002069** | -8.04E-06 | 6.32E-06 |
| age *vs* prior | -0.00017 | 0.00014 | **-0.00122** | **0.001975** | -1.02E-05 | 1.19E-05 |
| age *vs* gender | -0.00011 | 0.000135 | **-0.00178** | **0.001341** | -1.99E-05 | 1.68E-05 |
| age *vs* juvenilecrime | -0.00019 | 0.000139 | **-0.00167** | **0.002496** | -1.31E-05 | 8.15E-06 |
| age *vs* currentcharge | -0.0003 | 0.000351 | **-0.00257** | **0.00268** | -1.24E-05 | 1.61E-05 |

**FISC calculated from different sampling methods-AWP and random sampling**  Due to the limited availability of relevant sampling benchmarks in the current literature, we established an evaluation framework by considering two well-accepted methods as baselines: sampling from adversarial weight perturbations (AWP) Wu et al. (2020); Tsai et al. (2021) and sampling with different weight initialization seeds Li & Barnard (2022a); Semenova et al. (2019). Together with our introduced greedy algorithm, we utilized these methods to sample models from the Rashomon set. We adopt an MLP as the reference model and keep it consistent among all methods. With set $\epsilon = 0.05$, we constructed the Rashomon set and calculated the corresponding FISC, partial results presented in Table. 4. It is clear that our approach explored a broader range of FISC in comparison to the alternatives. Complete results are provided in Appendix I, with separate swarm plot visualizations.

**Going beyond the literature**  The logistic model from Dong & Rudin (2020) is not optimal so we train and improve the model as in Appendix H. We apply FISC to this model with settings $t = [0.2\epsilon, 0.4\epsilon, 0.6\epsilon, 0.8\epsilon, \epsilon]$, and the results are shown in Fig. 5(b). From the Fig. 5(b) we observe that the interaction of 'prior' and 'juvenile' or 'prior' and 'charge' can be extended to a similar extent. However, the compensation structure proposed by VIC fails to acknowledge the potential of the latter interaction. Additionally, we provide confirmation and explanation for their statement, as the latter case necessitates a greater loss change (indicated by a darker red color) compared to the

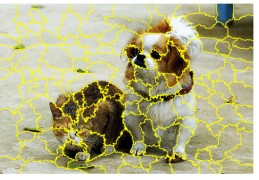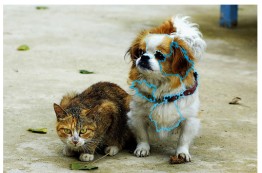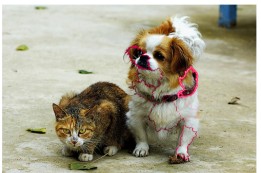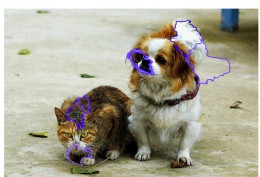

Figure 6: The utility of FISC in an image classification task. From left to right: the original segmented image, top 5 main effect segmentations for accurate dog predictions by the transformer (coloured in blue), top 5 important interacted segmentations for accurate dog predictions by the transformer (coloured in red), and top 5 important interacted segmentations for accurate cat predictions by the transformer (coloured in purple).

former, thereby facilitating easier compensation. We present the pairwise interaction and a triplewise interaction in Fig. 4, allowing us to observe the reliance on higher-order interactions. For instance, we observe that the decrease in reliance on the feature interaction 'prior' and 'juvenile' (x-y plane) can be compensated by the interaction between 'juvenile' and 'charge' (y-z plane) or 'prior' and 'charge' (x-z plane). Likewise, the color scheme depicted in the figure conveys equivalent information.

### 4.3 FEATURE INTERACTION IN IMAGE CLASSIFICATION WITH FISC

Over the past decade, large and pre-trained transformer architectures have achieved SOTA performance on a wide variety of tasks (Sarzynska-Wawer et al., 2021; Dosovitskiy et al., 2021). Here we aim to study how a large-scale model relies on feature interactions and how reasonably accurate predictions can differ in terms of their variances in FIS. We adopt the pre-trained SOTA vision transformer ViT-B-16 developed in Dosovitskiy et al. (2021); Paszke et al. (2019) with a top-5 accuracy of 95.32% on ImageNet and we chose an image from the COCO dataset that encompasses two objects, a cat and a dog, sharing similar visual characteristicsLin et al. (2014). The 3 primary predictions from the model classifying the image ranks as: *Pekinese* 0.49, *Japanese spaniel* 0.16, and *Tiger cat* 0.06. The model identifies the image as resembling a dog more than a cat.

The image is segmented as features by the quickshift method Vedaldi & Soatto (2008), following the protocol Tsang et al. (2020b) shown in Fig. 6(a). We use FISC to explain the interaction effects. The top 5 important segmentations are coloured blue in Fig. 6(b), where importance increases according to colour lightness. The most important part of the image is the forehead of the dog. A similar observation is reported in (Tsang et al., 2020b). Considering the joint effect resulting in the dog classification, the top 5 crucial pairwise interactions are given in Fig. 6(c). Notably, the lightest component interacts with other components more than once. Results shows that the model relies heavily on single forehead segmentation to predict a dog, but this segmentation is disappearing in the top 5 features for predicting a dog relying on interactions. If our objective is to classify the image as a cat, our findings suggest in Fig. 6(d) that the model would rely on both the dog and cat segmentations.

## 5 DISCUSSION

The motivation behind the development of FIS and FISC is to explain feature interaction in a model class, rather than focusing on a specific model. In this study, we highlight that feature interaction can vary significantly across models with similar performance. This work initializes the exploration of feature interaction in a Rashomon set. By leveraging the FISC, we approach the true feature interaction from a set of well-trained models. The variation of FIS allows for the explanation of diverse models that rely on distinct feature interactions and this property holds significant potential for selecting models that exhibit the desired feature interaction dependency, illustrated in synthetic validation, recidivism prediction and image classification with promising results. Further exploration of feature interaction with other ways of characterizing the Rashomon set is subject to future work. Additionally, we have developed a visualization tool that effectively illustrates the joint effect on the Rashomon set exploration and we adopt swarm plots to depict the variability of FISs in a model class. These tools has the potential to inspire future investigations in various domains. Our code is available at `https://github.com/Sichao-Li/generalized_rashomon_set`.

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

# A    MORE RELATED WORKS

An analysis of variance (ANOVA) test is employed to evaluate the significance of interaction terms in an additive model that includes all possible pairwise interactions Fisher (1992); Mandel (1961). The test examines the variance within each group compared to the variance between the groups and determines whether any differences observed in the means of the groups are likely due to chance or to a significant factor. H-statistics measure the strength of pairwise interactions based on the concept of Partial Dependence Function (PDF), which represents the relationship between a target variable and a specific feature while holding other features constant. The H-statistics can be calculated by comparing the PDF of the target feature with the PDF of the target feature conditioned on the presence of another feature. The method has been extended to feature importance and feature interaction detection Friedman & Popescu (2008); Greenwell et al. (2018). Sorokina et al. Sorokina et al. (2008) proposed a grove-based method called Additive Groves (AG) that extends the concept of decision tree ensembles to incorporate additive structure in the model. The method involves training individual trees to capture additive effects of the features on the target variable and then combining them to create a powerful ensemble model. By incorporating additivity into the model, Additive Groves can capture more complex relationships among features. In interaction detection, lasso-based methods are also widely used due to that they can quickly select interactions. One can construct an additive model with many different interaction terms and let lasso shrink the coefficients of unimportant terms to zero Bien et al. (2013).

Recently, more neural network-based methods are proposed to detect feature interactions. Bayesian Neural Networks (BNN)Cui et al. (2020) is to evaluate pairs of features with significant second-order derivatives at the input. By analyzing the posterior distribution of the model parameters, BNN can identify which pairs of features have a significant interaction effect. Specifically, the method evaluates the pairwise interaction by computing the difference between the posterior distributions of the model predictions with and without the interaction term. The Neural Interaction Detection (NID) method detects statistical interactions between features by examining the weight matrices of feed-forward neural networks Tsang et al. (2021). Specifically, interactions are determined by finding a cutoff on the ranking using a special form of the generalized additive model. Deep Feature Interaction Maps detect interactions between two features by calculating the change in the attribution of one feature incurred by changing the value of the second Greenside et al. (2018). It calculates the interaction between pairs of features by comparing the attribution maps generated when each feature is varied individually to the maps generated when both features are varied simultaneously. Singh et. al. Singh et al. (2019) generalized Contextual Decomposition Greenwell et al. (2018) to explain interactions for feed-forward and convolutional architectures.

Most recently, several works have been proposed to attribute predictions to feature interactions The Shapley-Taylor Interaction Index measures the contribution of pairwise feature interactions in machine learning models by combining the Shapley value and the Taylor expansion to estimate the interaction effects. The Shapley value quantifies the marginal contribution of each feature, while the Taylor expansion approximates the prediction function considering main effects and pairwise interactions Grabisch & Roubens (1999); Sundararajan et al. (2020). Integrated Hessians (IH) Janizek et al. (2021) is a method used to assess feature interactions in deep neural networks. It involves computing the integrated Hessian matrix and integrating the Hessian matrix over the input data, IH captures both the local and global curvature information of the loss landscape by analyzing the eigenvalues and eigenvectors of the integrated Hessian matrix. The Archipelago Tsang et al. (2020b) architecture is specifically designed for attributing feature interactions in machine learning models. It offers a framework based on mixed partial derivatives for identifying and quantifying the contributions of interactions between features. which consists of an interaction attribution method, ArchAttribute, and an interaction detector, ArchDetect.

These above methods show excellent accuracy in feature interaction predictions from a pre-specified model. However, as more researchers advocate exploring a set of equally good models, it is worthy discovering feature interactions in a model class, defined as the Rashomon set. The Rashomon set is named after Akira Kurosawa's film "Rashomon", and it refers to a collection of diverse but equally plausible models that achieve comparable performance on a given task or dataset. Fisher Fisher et al. (2019) introduces the concept of model class reliance (MCR) as a measure that captures the variability of Variable Importance (VI) values across a well-performing model class known as the Rashomon set. MCR provides a more comprehensive and nuanced understanding of feature

importance by considering the range of VI values obtained from multiple prediction models within the class. Following their work, Dong Dong & Rudin (2020) explores the cloud of variable importance, referred to as VIC for the set of all good models and provides concrete examples in linear regression and logistic regression. Another paper Li & Barnard (2022a) proposed a post-hoc method, variance tolerance factor (VTF) to interpret a set of neural networks by greedy searching all possible neural networks with certain conditions.

## B  UNVEILING THE MASK-BASED SAMPLING RASHOMON SET

As stated that it's always easier to find an accurate but complex model, it refers to a model with more complex structures. It's safely claimed that concatenating layers into a backbone model increases the complexity and therefore we logically demonstrated that such a model with more mask layers can achieve similar or better performance. Then we applied propositional logic to prove, for any model in the Rashomon set, we can find an alternative by adding mask layers into the reference model (same as the backbone model). Our aim can be mathematically represented as: $\forall f \in \mathcal{F} \left(\mathbb{E}[L(f(\boldsymbol{X}), \boldsymbol{y})] \rightarrow (\exists m \left(\mathbb{E}[L(f^* \circ m(\boldsymbol{X}), \boldsymbol{y})] \simeq \mathbb{E}[L(f(\boldsymbol{X}), \boldsymbol{y})]\right)\right)$ with details below:

| | | |
|---|---|---|
| 1 | $\forall f \in \mathcal{F} \left(\mathbb{E}[L(f(\boldsymbol{X}), \boldsymbol{y})] \rightarrow \mathbb{E}[L(f(\boldsymbol{X}), \boldsymbol{y})] \leq \mathbb{E}L(f^*(\boldsymbol{X}), \boldsymbol{y}) + \epsilon\right)$ | |
| 2 | $\forall f \in \mathcal{F} \left(\mathbb{E}[L(f(\boldsymbol{X}), \boldsymbol{y})] \rightarrow \exists m \left(\mathbb{E}[L(f \circ m(\boldsymbol{X}), \boldsymbol{y})] \leq \mathbb{E}[L(f(\boldsymbol{X}), \boldsymbol{y})]\right)\right)$ | |
| 3 | $f'$ $\left(\mathbb{E}[L(f'(\boldsymbol{X}), \boldsymbol{y})] \rightarrow \mathbb{E}[L(f'(\boldsymbol{X}), \boldsymbol{y})] \leq \mathbb{E}[L(f^*(\boldsymbol{X}), \boldsymbol{y})] + \epsilon\right)$ | |
| 4 | $\left(\mathbb{E}[L(f'(\boldsymbol{X}), \boldsymbol{y})] \rightarrow \exists m \left(\mathbb{E}[L(f' \circ m(\boldsymbol{X}), \boldsymbol{y})] \leq \mathbb{E}[L(f'(\boldsymbol{X}), \boldsymbol{y})]\right)\right)$ | |
| 5 | $\mathbb{E}[L(f'(\boldsymbol{X}), \boldsymbol{y})]$ | |
| 6 | $\mathbb{E}[L(f'(\boldsymbol{X}), \boldsymbol{y})] \leq \mathbb{E}[L(f^*(\boldsymbol{X}), \boldsymbol{y})] + \epsilon)$ | $\rightarrow$-E, 3, 5 |
| 7 | $\exists m \left(\mathbb{E}[L(f^* \circ m(\boldsymbol{X}), \boldsymbol{y})] \leq \mathbb{E}[L(f^*(\boldsymbol{X}), \boldsymbol{y})]\right)$ | $\rightarrow$-E, 4, 5 |
| 8 | $m'$ $\mathbb{E}[L(f^* \circ m'(\boldsymbol{X}), \boldsymbol{y})] \simeq \mathbb{E}[L(f^*(\boldsymbol{X}), \boldsymbol{y})]$ | |
| 9 | $\mathbb{E}[L(f'(\boldsymbol{X}), \boldsymbol{y})] \leq \mathbb{E}[L(f^* \circ m'(\boldsymbol{X}), \boldsymbol{y})] + \epsilon$ | R, 6, 8 |
| 10 | $\mathbb{E}[L(f'(\boldsymbol{X}), \boldsymbol{y})] \leq \mathbb{E}[L(f'(\boldsymbol{X}), \boldsymbol{y})] + \epsilon$ | Logic, 6, 8 |
| 11 | $\exists m \, \mathbb{E}[L(f^* \circ m(\boldsymbol{X}), \boldsymbol{y})] \simeq \mathbb{E}L(f'(\boldsymbol{X}), \boldsymbol{y})$ | $\exists$-I, 9, 10 |
| 12 | $\exists m \left(\mathbb{E}[L(f^* \circ m(\boldsymbol{X}), \boldsymbol{y})] \simeq \mathbb{E}L(f'(\boldsymbol{X}), \boldsymbol{y})\right)$ | $\exists$-E, 7, 8–11 |
| 13 | $\mathbb{E}[L(f'(\boldsymbol{X}), \boldsymbol{y})] \rightarrow \exists m \left(\mathbb{E}L(f^* \circ m(\boldsymbol{X}), \boldsymbol{y}) \simeq \mathbb{E}[L(f'(\boldsymbol{X}), \boldsymbol{y})]\right)$ | $\rightarrow$-I, 5–12 |
| 14 | $\forall f \in \mathcal{F} \left(\mathbb{E}[L(f(\boldsymbol{X}), \boldsymbol{y})] \rightarrow (\exists m \left(\mathbb{E}[L(f^* \circ m(\boldsymbol{X}), \boldsymbol{y})] \simeq \mathbb{E}[L(f(\boldsymbol{X}), \boldsymbol{y})]\right)\right)$ | $\forall$-I, 3, 13 |

## C  ADDITIONAL PROOF OF CONCEPT

### C.1  PROBLEM SETTING

To illustrate the idea, we designed an additional experiment in the synthetic dataset and solved the quadratic problem. The problem is to train a model to solve the quadratic equation $ax^2 + bx + c = 0$, where the variables $a$, $b$, and $c$ are inputs and $x_1$ and $x_2$ are the outputs. Based on mathematical principles, two error-free models can be found as $x = \frac{-b \pm \sqrt{b^2 - 4ac}}{2a}$.

Given the function, we randomly sampled 12,000 data points from the uniform distribution $a, b, c \sim \mathcal{U}(0, 1)$ with fixed seed as input and calculated outputs accordingly. It's noted that the outputs might be complex numbers. The train/test/validation set is split as 0.8/0.1/0.1. The regression problem can be fit using different types of models. Here we applied a traditional artificial neural network and achieved MSE 0.3353 in the training set.

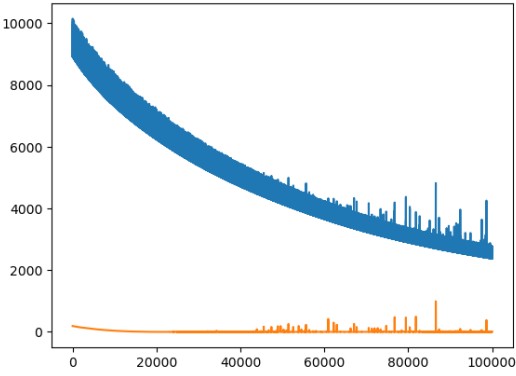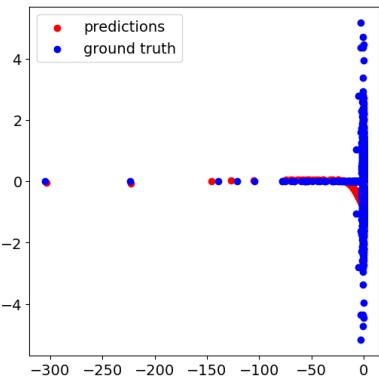

Figure 7: Left: Learning curve. Right: Prediction VS Ground Truth.

## C.2 GROUND TRUTH INTERACTIONS

The given function $f^* = \frac{-b \pm \sqrt{b^2 - 4ac}}{2a}$ can be seen as an ideal model with $\mathbb{E}[L(f^*(\boldsymbol{X}))] = 0$, which corresponds to a true feature interaction. The expected FIS of $a, b$ can be calculated as follows:

$$
\begin{aligned}
FIS_{a,b}(f^*) =& \varphi_{a,b}(f^*) - (\varphi_a(f^*) + \varphi_b(f^*)) \\
=& \mathbb{E}[L(f^*(\boldsymbol{X}_{\backslash\{a,b\}}))] - \mathbb{E}[L(f^*(\boldsymbol{X}_{\backslash\{a\}}))] - \mathbb{E}[L(f^*(\boldsymbol{X}_{\backslash\{b\}}))] \\
\simeq& \sum_{i=1}^n (f^*(\boldsymbol{X}_{\backslash\{a,b\}}) - \boldsymbol{y})^2 - \sum_{i=1}^n (f^*(\boldsymbol{X}_{\backslash\{a\}}) - \boldsymbol{y})^2 - \sum_{i=1}^n (f^*(\boldsymbol{X}_{\backslash\{b\}}) - \boldsymbol{y})^2
\end{aligned}
\tag{10}
$$

Where $n$ is the number of samples and interaction between $b, c$ and $a, c$ can be calculated in the same way. The interaction values are absolute to indicate the strength. The interaction values are absolute to indicate the strength, summarized in the Table. 5.

Table 5: Ground truth of feature importance and interaction

| $\varphi_a(f^*)$ | $\varphi_b(f^*)$ | $\varphi_c(f^*)$ | $\varphi_{a,b}(f^*)$ | $\varphi_{a,c}(f^*)$ | $\varphi_{b,c}(f^*)$ |
|---|---|---|---|---|---|
| 116.4602 | 24.2023 | 0.2924 | 21.2866 | 1.3813 | 0.5945 |

## C.3 APPROXIMATED FISC

Now, assume that we do not have an optimal model in the real-world and our objective is to train a model to fit the data, the regression problem is fitted using a simple neural network and achieved MSE=0.33 in the test set. The learning curve and prediction versus ground truth are illustrated in Fig. 7. After training the model, we applied the FISC to it and plotted the swarm plots, which are provided in Fig. 8. Each point in the swarm plot represents an FIS value from a specific model.

when we consider the leftmost point among the feature pair a, b, it implies that there exists *a model with a lower FIS for a, b compared to a, c*. This means that there was a model trained that happened to have a lower FIS for a, b than b, c using the same dataset with promising loss. This observation contradicts the truth interaction calculated previously. However, through the analysis of FISC, we found that the range of FIS values for each feature pair covers the truth interaction. By examining the statistics of the results, we can approach true feature interactions, which will be the focus of our further work.

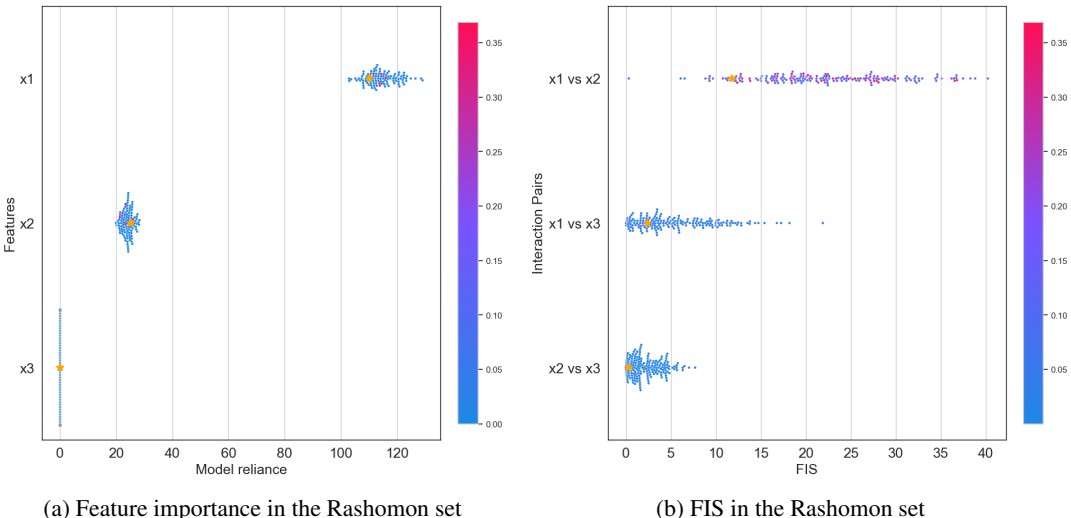

(a) Feature importance in the Rashomon set

(b) FIS in the Rashomon set

Figure 8: Results from a set of models sampled from the well-trained neural network.

## D  RELATION TO ARCHDETECT

The framework ArchDetect Tsang et al. (2020b) uses $\frac{\delta}{h_i h_j}$ to detect the interaction existence $\delta$ defined as

$$\delta = f(\boldsymbol{x}^*_{\{i,j\}} + \boldsymbol{x}_{\setminus\{i,j\}}) + f(\boldsymbol{x}'_{\{i,j\}} + \boldsymbol{x}_{\setminus\{i,j\}})$$
$$- f(\boldsymbol{x}'_{\{i\}} + \boldsymbol{x}^*_{\{j\}} + \boldsymbol{x}_{\setminus\{i,j\}}) - f(\boldsymbol{x}'_{\{j\}} + \boldsymbol{x}^*_{\{i\}} + \boldsymbol{x}_{\setminus\{i,j\}}) \qquad (11)$$

based on mixed partial derivatives, where $f$ is a black box model, $\boldsymbol{x}^*$ represents the sample we want to explain and $\boldsymbol{x}'$ represents a neutral reference sample, which is often zero vectors. If $\delta > 0$, then interaction exists, otherwise not. The author claimed $100\%$ accuracy in a synthetic dataset and defined the feature interaction strength as the square of $\frac{\delta}{h_i h_j}$. Here we show that for any model $\mathcal{M}_{\boldsymbol{m}}$ in the model class, the FIS value is equivalent to $\delta$ under conditions that the loss is set as RMSE. In our setting, for any model $\mathcal{M}_{\boldsymbol{m}}$, the expected RMSE loss is $\mathbb{E}[L(\mathcal{M}_{\boldsymbol{m}_{\mathcal{I}}}(\boldsymbol{X}), \boldsymbol{y})] = \mathbb{E}[\sqrt{(\boldsymbol{y} - \mathcal{M}_{\boldsymbol{m}_{\mathcal{I}}}(\boldsymbol{X}))^2}]$. For simplicity, we consider $\mathbb{E}[\boldsymbol{y} - \mathcal{M}_{\boldsymbol{m}_{\mathcal{I}}}(\boldsymbol{X})]$ (meaning assuming the model approximate the true solutions from either below or above). For $\mathcal{I} = \{i, j\}$ and $\forall \mathcal{M}_{\boldsymbol{m}_{ij}} \in \mathcal{R}(\epsilon, g, \mathcal{F})$, following the definition, we can derive

$$FIS_{\mathcal{I}}(\mathcal{M}_{\boldsymbol{m}_{\mathcal{I}}}) = (\mathbb{E}[\mathcal{M}_{\boldsymbol{m}_{i,j}}(\boldsymbol{X}_{\setminus\{i,j\}})] + \mathbb{E}[\mathcal{M}_{\boldsymbol{m}_{i,j}}(\boldsymbol{X})]$$
$$- \mathbb{E}[\mathcal{M}_{\boldsymbol{m}_{ij}}(\boldsymbol{X}_{\setminus\{i\}})] - \mathbb{E}[\mathcal{M}_{\boldsymbol{m}_{i,j}}(\boldsymbol{X}_{\setminus\{j\}})]), \qquad (12)$$

which is mathematically equivalent to (11) in view of the decomposition of the interaction into four terms.

## E  DERIVATION OF THE UPPER AND LOWER BOUND OF FISC IN MLP

### E.0.1  CHARACTERIZING THE MASK-BASED RASHOMON SET

We first apply the mask $\boldsymbol{m} = \{m_1, m_2, ..., , m_p\}$ to the reference model as $\mathcal{M}_{\boldsymbol{m}} = f \circ \boldsymbol{m}$. To characterize the mask-based Rashomon set, the goal is to find the conditions for $\boldsymbol{m}$ such that:

$$\mathbb{E}[L(\mathcal{M}_{\boldsymbol{m}}(\boldsymbol{X}))] \leq \mathbb{E}[L(f(\boldsymbol{X}))] + \epsilon. \qquad (13)$$

After applying the above settings, we can rewrite Eq. 13 as:

$$\mathbb{E}[\boldsymbol{y} - \boldsymbol{\alpha}^T \frac{1}{1 + e^{-\boldsymbol{\beta}^T \boldsymbol{X}}} + b] - \epsilon \leq \mathbb{E}[\boldsymbol{y} - \boldsymbol{\alpha}^T \frac{1}{1 + e^{-\boldsymbol{m}^T \cdot \boldsymbol{\beta}^T \boldsymbol{X}}} + b] \leq \mathbb{E}[\boldsymbol{y} - \boldsymbol{\alpha}^T \frac{1}{1 + e^{-\boldsymbol{\beta}^T \boldsymbol{X}}} + b] + \epsilon.$$

Without loss of generality, we assume $\boldsymbol{\alpha} > 0$ and a scaling on $\epsilon$ by $\boldsymbol{\alpha}$. This can be simplified to

$$\mathbb{E}[\frac{1}{1+e^{-\boldsymbol{\beta}^T\boldsymbol{X}}}] - \epsilon \leq \mathbb{E}[\frac{1}{1+e^{-\boldsymbol{m}^T\cdot\boldsymbol{\beta}^T\boldsymbol{X}}}] \leq \mathbb{E}[\frac{1}{1+e^{-\boldsymbol{\beta}^T\boldsymbol{X}}}] + \epsilon. \tag{14}$$

With further assumption[4] that $\epsilon \leq \min\{\mathbb{E}[\frac{1}{1+e^{-\boldsymbol{m}^T\cdot\boldsymbol{\beta}^T\boldsymbol{X}}}], \mathbb{E}[\frac{e^{-\boldsymbol{m}^T\cdot\boldsymbol{\beta}^T\boldsymbol{X}}}{1+e^{-\boldsymbol{m}^T\cdot\boldsymbol{\beta}^T\boldsymbol{X}}}]\}$, this inequality is reduced to

$$\mathbb{E}\left[\ln\left(\frac{1-\epsilon-\epsilon e^{-\boldsymbol{\beta}^T\boldsymbol{X}}}{e^{-\boldsymbol{\beta}^T\boldsymbol{X}}+\epsilon+\epsilon e^{-\boldsymbol{\beta}^T\boldsymbol{X}}}\right)\right] \leq \mathbb{E}[(\boldsymbol{\beta}^T\boldsymbol{X})\boldsymbol{m}^T] \leq \mathbb{E}\left[\ln\left(\frac{1+\epsilon+\epsilon e^{-\boldsymbol{\beta}^T\boldsymbol{X}}}{e^{-\boldsymbol{\beta}^T\boldsymbol{X}}-\epsilon-\epsilon e^{-\boldsymbol{\beta}^T\boldsymbol{X}}}\right)\right], \tag{15}$$

which gives the condition on $\boldsymbol{m}$ that characterises the mask-based Rashomon set.

### E.0.2 Calculating the expected FISC lower and upper bounds

With the above setting, we calculate the FIS:

$$FIS_{\mathcal{I}}(\mathcal{M}_{\boldsymbol{m}}) = \mathbb{E}[\boldsymbol{y} - (\boldsymbol{\alpha}^T\frac{1}{1+e^{-\boldsymbol{m}^T\cdot\boldsymbol{\beta}^T\boldsymbol{X}_{\backslash i,j}}} + b)] - \mathbb{E}[\boldsymbol{y} - (\boldsymbol{\alpha}^T\frac{1}{1+e^{-\boldsymbol{m}^T\cdot\boldsymbol{\beta}^T\boldsymbol{X}}} + b)]-$$

$$(\mathbb{E}[\boldsymbol{y} - (\boldsymbol{\alpha}^T\frac{1}{1+e^{-\boldsymbol{m}^T\cdot\boldsymbol{\beta}^T\boldsymbol{X}_{\backslash i}}} + b)] - \mathbb{E}[\boldsymbol{y} - (\boldsymbol{\alpha}^T\frac{1}{1+e^{-\boldsymbol{m}^T\cdot\boldsymbol{\beta}^T\boldsymbol{X}}} + b)]+$$

$$\mathbb{E}[\boldsymbol{y} - (\boldsymbol{\alpha}^T\frac{1}{1+e^{-\boldsymbol{m}^T\cdot\boldsymbol{\beta}^T\boldsymbol{X}_{\backslash j}}} + b)] - \mathbb{E}[\boldsymbol{y} - (\boldsymbol{\alpha}^T\frac{1}{1+e^{-\boldsymbol{m}^T\cdot\boldsymbol{\beta}^T\boldsymbol{X}}} + b)]). \tag{16}$$

To find the minimum and maximum values of FIS, we apply the condition (domain) Eq. 6 and the critical point (when $\boldsymbol{m} = \boldsymbol{m}_\star$ such that $\frac{FIS_{\mathcal{I}}(\mathcal{M}_{\boldsymbol{m}})}{\partial\boldsymbol{m}} = 0$) of the function $FIS_{\mathcal{I}}(\mathcal{M}_{\boldsymbol{m}})$. We denote

$$\boldsymbol{m}_1 = \frac{1}{\boldsymbol{\beta}^T\boldsymbol{X}}\ln\left(\frac{1-\epsilon-\epsilon e^{-\boldsymbol{\beta}^T\boldsymbol{X}}}{e^{-\boldsymbol{\beta}^T\boldsymbol{X}}+\epsilon+\epsilon e^{-\boldsymbol{\beta}^T\boldsymbol{X}}}\right), \quad \boldsymbol{m}_2 = \frac{1}{\boldsymbol{\beta}^T\boldsymbol{X}}\ln\left(\frac{1+\epsilon+\epsilon e^{-\boldsymbol{\beta}^T\boldsymbol{X}}}{e^{-\boldsymbol{\beta}^T\boldsymbol{X}}-\epsilon-\epsilon e^{-\boldsymbol{\beta}^T\boldsymbol{X}}}\right). \tag{17}$$

Now, one can denote

$$FIS_{\min} := \inf_{\boldsymbol{m}} FIS_{\mathcal{I}}(\mathcal{M}_{\boldsymbol{m}}) = \min\{FIS_{\mathcal{I}}(\mathcal{M}_{\boldsymbol{m}_1}), FIS_{\mathcal{I}}(\mathcal{M}_{\boldsymbol{m}_2}), FIS_{\mathcal{I}}(\mathcal{M}_{\boldsymbol{m}_\star})\},$$
$$FIS_{\max} := \sup_{\boldsymbol{m}} FIS_{\mathcal{I}}(\mathcal{M}_{\boldsymbol{m}}) = \max\{FIS_{\mathcal{I}}(\mathcal{M}_{\boldsymbol{m}_1}), FIS_{\mathcal{I}}(\mathcal{M}_{\boldsymbol{m}_2}), FIS_{\mathcal{I}}(\mathcal{M}_{\boldsymbol{m}_\star})\}. \tag{18}$$

The range of FISC is then characterized as $[FISC_{\mathcal{I}}(\mathcal{R})_{min}, FISC_{\mathcal{I}}(\mathcal{R})_{max}] = [FIS_{\min}, FIS_{\max}]$.

**Remark.** *The exact critical point $\boldsymbol{m} = \boldsymbol{m}_\star$ such that $\frac{FIS_{\mathcal{I}}(\mathcal{M}_{\boldsymbol{m}})}{\partial\boldsymbol{m}} = 0$ is difficult to obtain as it requires solutions involving a polynomial of degree seven and exponential/logarithmic functions. However, this can be approximately by root-finding algorithms such as Newton's method. Another approximation is to use a first-order Taylor expansion of $FIS_{\mathcal{I}}(\mathcal{M}_{\boldsymbol{m}})$ at $\boldsymbol{m} = 1$. The analytical expression is still extremely complicated, posing difficulties in finding extreme values of FIS, so we obtain this critical point by a root-finding algorithm. We present a generic method in Sec. 3.2.*

## F Higher order interaction visualization discussion

In our paper, we provide visualizations of pairwise interaction in halo plot and swarm plot, and triplewise interaction in halo plot. These two-level interactions are the most studied in the literature and higher-order interactions are sometimes considered redundant in terms of attribution (Bien et al., 2013; Dong & Rudin, 2020). Theoretically, halo plot and swarm plot can visualize any order of interactions and swarm plots can visualize any order of interactions empirically. However, visualizing higher-dimensional spaces (>3) can be challenging due to the limitations of human perception, which makes directly visualizing higher-order interactions using halo plots not feasible.

We acknowledge this limitation and offer suggestions for users who insist visualizing higher-order interactions (>3) using halo plots. To visualize higher-order interactions (>3) in halo plots, we can apply some commonly used higher-order visualization methods, e.g., encoding color information as 4th dimension and applying dimensionality reduction. To demonstrate this, we provided an example of visualizing 4-way interactions. The experiment is based on the recidivism prediction in Sec. 4. On the existence of 3-way interaction visualization, we encoded color as 4th feature, where lighter colors indicate the larger interactions, as shown in Fig. 9.

---

[4] A smaller $\epsilon$ means a smaller Rashomon set with models closer to the reference well-trained model $f_{ref}$.

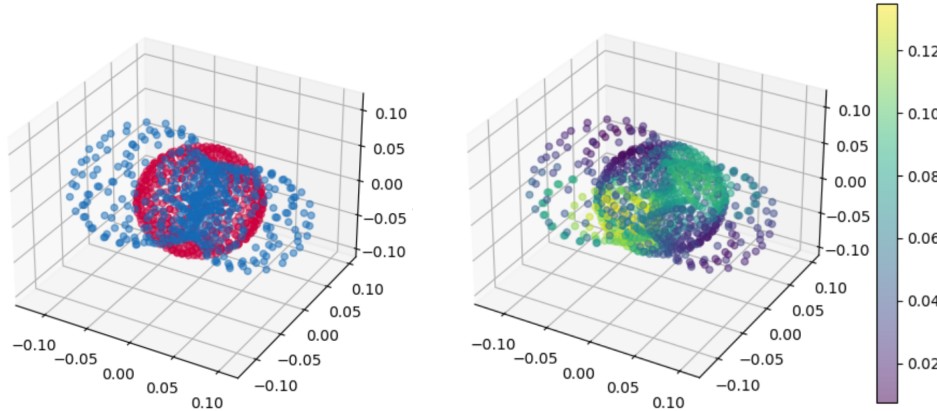

Figure 9: Left: 3-D halo plot. Right: 4-D halo plot

## G   ILLUSTRATION OF HALO PLOT WITH A CONCRETE EXAMPLE

Following the setting in the main text that function is $f = x_i + x_j + x_i * x_j$ and $\epsilon = 0.1$. We choose a scenario when the main effect of feature $x_i$ is $\phi_i = 0.01$ and the main effect of feature $x_j$ is $\phi_j = 0.09$. Given $\phi_i = 0.01$, we can derive the following with Mean Squared Error (MSE) loss function

$$
\begin{aligned}
\phi_i &= L(f \circ m_i(X) - L(f(X)) \\
&= (y - (m_i x_i + x_j + m_i x_i * x_j))^2 - (y - (x_i + x_j + x_i * x_j))^2 \\
&= 0.01.
\end{aligned} \tag{19}
$$

Similarly,

$$
\begin{aligned}
\phi_j &= L(f \circ m_j(X) - L(f(X)) \\
&= (y - (x_i + m_j x_j + x_i * m_j x_j))^2 - (y - (x_i + x_j + x_i * x_j))^2 \\
&= 0.09.
\end{aligned} \tag{20}
$$

Considering variables sampling from the normal distribution $x_i \sim \mathcal{N}(\mu, \sigma^2) \, ; x_j \sim \mathcal{N}(\mu, \sigma^2)$ and $y$ is ground truth from the function. We can determine two possible values for $m_i$, namely $0.95$ and $1.05$, as well as two possible values for $m_j$, namely $0.85$ and $1.15$. From these options, we can identify four potential sets that guarantee the sum of main effects to be $\epsilon = 0.1$: $(m_i, m_j) = (0.95, 0.85), (0.95, 1.15), (1.05, 0.85), (1.05, 1.15)$. Next, we proceed to compute $\phi_{i,j} = (y - (m_i x_i + m_j x_j + m_i x_i * m_j x_j))^2 - (y - (x_i + x_j + x_i * x_j))^2$ using the aforementioned sets. These calculations allow us to generate four values of $\tilde{\epsilon}$, which can be plotted along a circle with a radius of $\epsilon$. Following the above procedure, we collected 9 ordered pairs $\{(\phi_i * x, \phi_j * y) \mid (x, y) \in \mathbb{R}, 0.1 \leq x, y \leq 0.9, x + y = 1\}$ and plotted 36 points, shown in Fig. 10.

## H   OPTIMIZING THE REGRESSOR FROM VIC

We downloaded the public code from `https://zenodo.org/record/4065582#.ZGIHVHZBzHI` and extracted model parameters from the logistic model in VIC. We realized that the model can be optimized further and showed the comparison between two models in loss vs individual masks of each feature in Fig. 11.

## I   COMPLETE RESULTS FROM THREE SAMPLING METHODS

Due to the page limitation, here we provided the complete results from three sampling methods: sampling from adversarial weight perturbations (AWP), sampling with different weight initialization seeds and greedy search approach, summarized in Table 6. To provide a more comprehensive

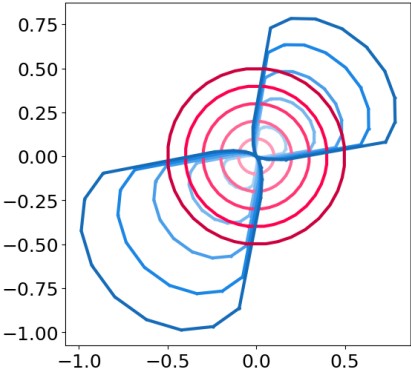

Figure 10: Exploring feature interaction of function $f = x_i + x_j + x_i * x_j$ in the Rashomon set. From inner to outer, the radii are 0.1, 0.2, 0.3, 0.4, and 0.5, respectively.

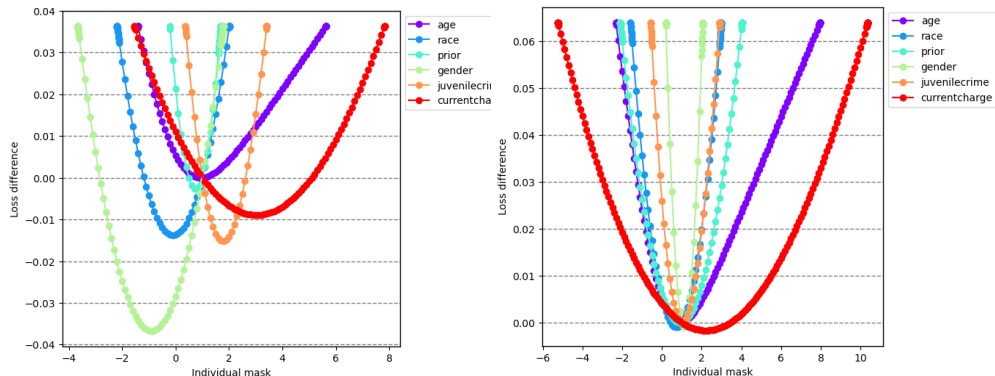

Figure 11: Left: greedy search algorithm applied in a logistic classifier extracted from VIC Dong & Rudin (2020). Right: greedy algorithm applied in another optimal logistic classifier. The x-axis refers to individual masks while the y-axis represents the loss change from the base model.

understanding of FISC, swarm plots are displayed below. In these plots, each data point represents a FIS derived from the sampled models, without absolute value transformation, as this comparison more directly illustrates the distribution of FISC across different sampling methods.

## J AN APPLICATION IN REAL-WORLD DATASET

To illustrate the usage of FISC, we applied our method to MXenes, an early transition metal carbides with two-dimensional (2D) structures exhibiting metallic conductivity and hydrophilicity. These materials are denoted by $M_{n+1}X_nT_x$, with $M$ representing an early transition metal such as Sc or Ti, $X$ signifying either Carbon (C) or Nitrogen (N) and functional groups $T$ (e.g., O, F, OH) for surface termination Gogotsi & Anasori (2019); Gogotsi & Huang (2021). By intercalating different ions and molecules like $Li^+$ and $K^+$, MXenes' electrochemical attributes such as voltage, induced charge and capacity can be modulated, serving as a potential battery material.

We used a dataset that contains 360 compounds intercalated with $Li^+$, $Na^+$, $K^+$, and $Mg^{2+}$ ions from Eames & Islam (2014); Li & Barnard (2022b). We represented the dataset by their categories, for instance, category Z encoded by 0–3 to describe Li, Na, K, and Mg, respectively, aiming to discover the interactions between structures. The dataset is split 90/10 for training/testing. We trained 3 MLPs for predicting their 3 properties: voltage, induced charge and capacity, individually. The $R^2$ for each reference model on testing set is reported in Figs. 15, 16, 17. We searched the Rashomon set based on the well-trained reference models with $\epsilon = 0.05$ and corresponding swarm plots are provided in Figs. 15, 16, 17.

Table 6: Comparison of FISC calculated from different sampling methods (complete). We use *vs* to denote the interaction pairs and best results are highlighted in bold.

| Interaction pairs | AWP sampling $FIS_{min}$ | $FIS_{max}$ | Greedy sampling $FIS_{min}$ | $FIS_{max}$ | Random sampling $FIS_{min}$ | $FIS_{max}$ |
|---|---|---|---|---|---|---|
| age *vs* race | -0.00026 | 0.000113 | **-0.00089** | **0.002069** | -8.04E-06 | 6.32E-06 |
| age *vs* prior | -0.00017 | 0.00014 | **-0.00122** | **0.001975** | -1.02E-05 | 1.19E-05 |
| age *vs* gender | -0.00011 | 0.000135 | **-0.00178** | **0.001341** | -1.99E-05 | 1.68E-05 |
| age *vs* juvenilecrime | -0.00019 | 0.000139 | **-0.00167** | **0.002496** | -1.31E-05 | 8.15E-06 |
| age *vs* currentcharge | -0.0003 | 0.000351 | **-0.00257** | **0.00268** | -1.24E-05 | 1.61E-05 |
| race *vs* prior | -0.00029 | 0.000225 | **-0.00151** | **0.006728** | -1.23E-05 | 1.81E-05 |
| race *vs* gender | -0.0002 | 0.000158 | **-0.00224** | **0.000606** | -9.43E-06 | 1.34E-05 |
| race *vs* juvenilecrime | -0.00019 | 0.000189 | **-0.00851** | **0.001587** | -2.13E-05 | 1.13E-05 |
| race *vs* currentcharge | -0.00034 | 0.00027 | **-0.00719** | **0.002191** | -1.87E-05 | 1.43E-05 |
| prior *vs* gender | -0.00018 | 0.000191 | **-0.0006** | **0.002526** | -1.58E-05 | 2.50E-05 |
| prior *vs* juvenilecrime | -0.00019 | 0.00022 | **-0.0068** | **0.002592** | -2.06E-05 | 1.91E-05 |
| prior *vs* currentcharge | -0.00029 | 0.00022 | **-0.00423** | **0.002219** | -2.23E-05 | 2.16E-05 |
| gender *vs* juvenilecrime | -0.00012 | 0.000148 | **-0.00828** | **0.000754** | -1.24E-05 | 1.04E-05 |
| gender *vs* currentcharge | -0.00035 | 0.000211 | **-0.01362** | **0.003708** | -2.01E-05 | 1.61E-05 |
| juvenilecrime *vs* currentcharge | -0.00022 | 0.000498 | **-0.00917** | **0.000693** | -2.14E-05 | 2.28E-05 |

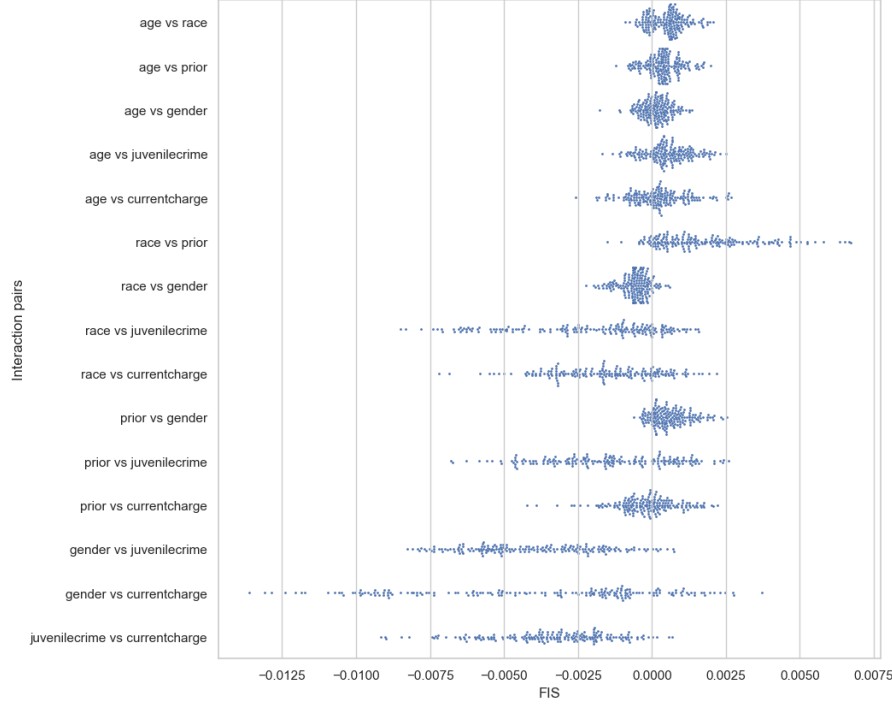

Figure 12: FISC from our greedy sampling method

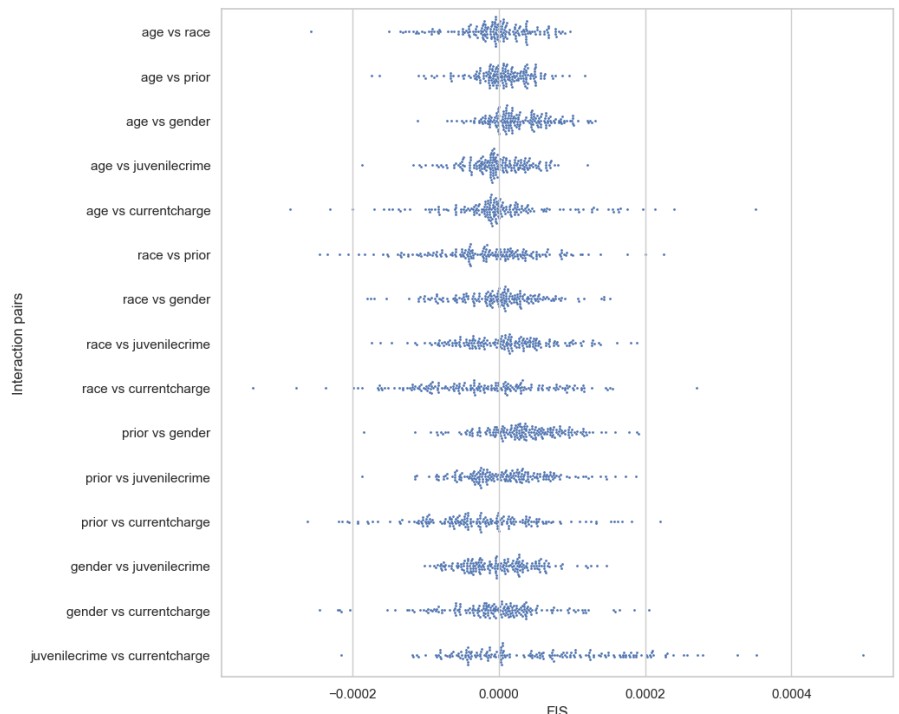

Figure 13: FISC from AWP sampling method

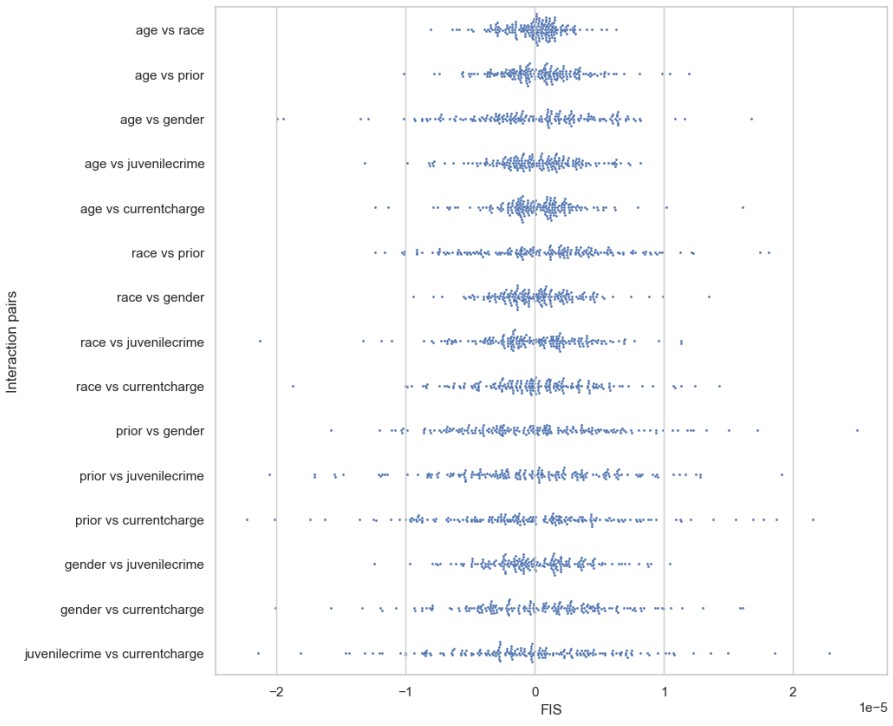

Figure 14: FISC from random sampling method

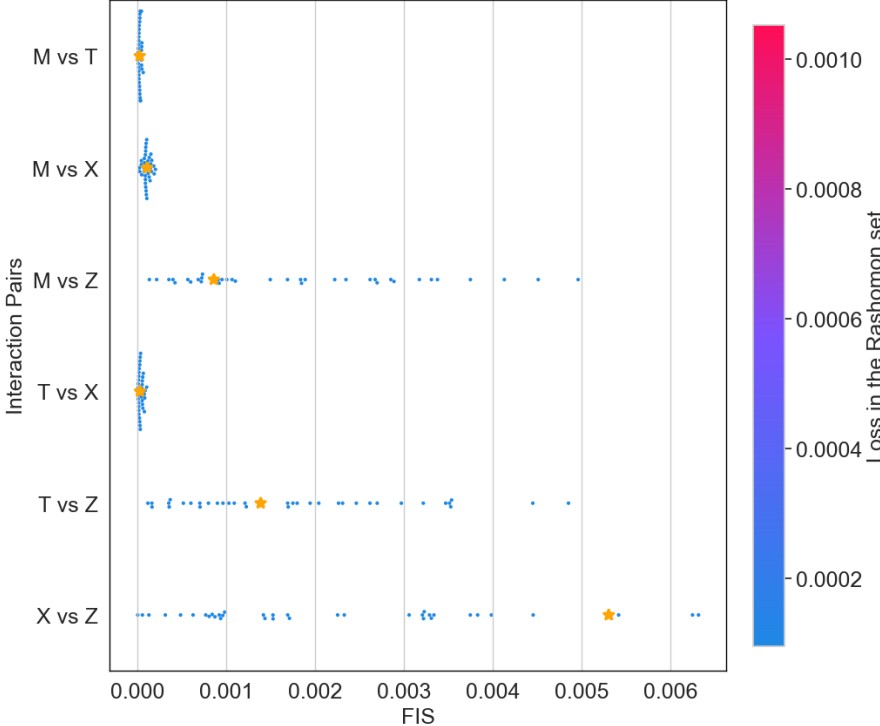

Figure 15: FISC from charge prediction. The $R^2$ of the reference model (depicted by the yellow star) achieves 0.98 on the testing set.

The results reveal a noteworthy observation: FIS exhibits a broad range when category Z is involved in interactions during the prediction of induced charge. This aligns with our common knowledge, given that ions, which form the category Z, are the cause of induced charge. These insightful results offer a more comprehensive understanding of feature interactions within the Rashomon set and potentially guide researchers in making informed decisions for future research.

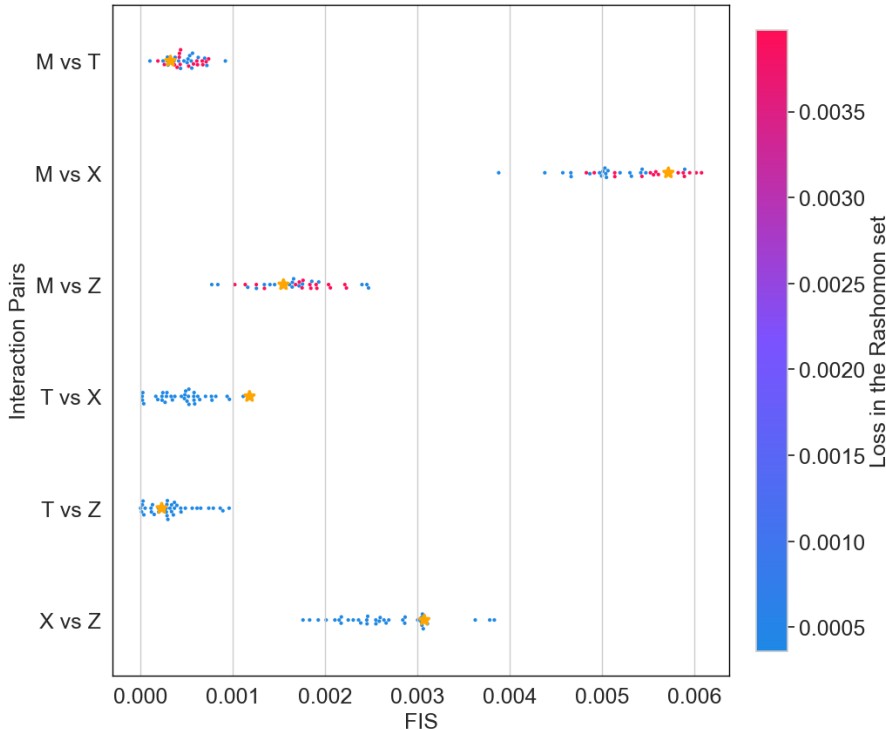

Figure 16: FISC from voltage prediction. The $R^2$ of the reference model (depicted by the yellow star) achieves 0.71 on the testing set.

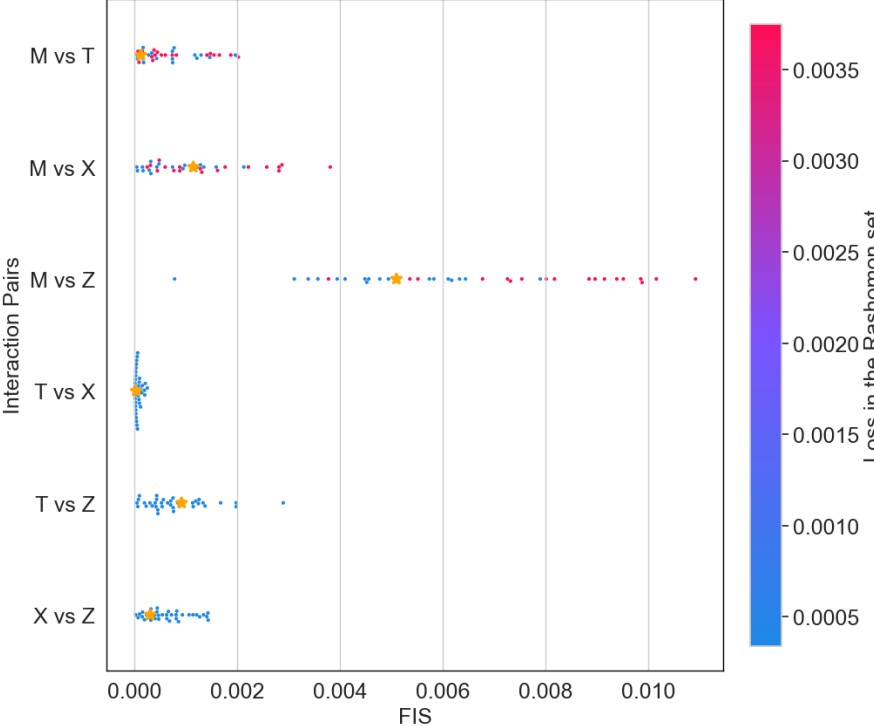

Figure 17: FISC from capacity prediction. The $R^2$ of the reference model (depicted by the yellow star) achieves 0.90 on the testing set.

