# OpenReview forum: "Exploring the cloud of feature interaction scores in a Rashomon set"
_ICLR.cc/2024/Conference — ICLR 2024 poster_

### Official Review · Reviewer_UrRA · 2023-10-30

**Soundness:** 3 good
**Presentation:** 2 fair
**Contribution:** 3 good
**Rating:** 6
**Confidence:** 3

**Summary:**

This paper argues that a well-trained predictive model may not accurately preserve the true feature interactions, and multiple well-performing predictive models can exhibit variations in feature interaction strengths. Therefore, they recommend exploring feature interaction strengths within a model class consisting of approximately equally accurate predictive models. The authors suggest exploring feature interaction strengths within a model class comprising models that are approximately equally accurate. The authors introduce the concept of Feature Interaction Score (FIS) within the framework of a Rashomon set. To facilitate the calculation of the FIS within this model class, they present a practical algorithm to calculate FIS. FIS is a straightforward and heuristic method, but it is still novel to me.

**Strengths:**

1. The originality of this paper is great. The problem is clearly defined.

2. The proposed method is technically sound to me.

3. The definition of FIS is a novel and reasonable tool to analyze feature interactions intuitively.

**Weaknesses:**

1. The overall presentations need to be improved. Many figures do not have the axes' labels.
2. The experiments should be conducted on a broader range of datasets.

**Questions:**

1. Is your work the first one to propose to use the loss change to measure the strength of feature interactions?
2. From my perspective,  feature interactions typically don't have a definitive "ground truth." Therefore, how can you prove the superiority of your Feature Interaction Score (FIS) compared to other baselines without directly comparing it to a ground truth?
3. What is MCR in Table 3?
4. What is the range of mask $m_i$? Mask values are used to be 0-1. However, in your algorithm, they seem to be $m_i\in \mathbb{R}$.

---

> ### Author Response · Authors · 2023-11-15
>
> I greatly appreciate your support for our work and characterizing it as original and novel. Here are our responses to your proposed questions and suggestions:
>
>
>
> ---
>
>
>
> 1. **The overall presentations need to be improved. Many figures do not have the axes' labels.**
>
> - Following your suggestions, we have added axes' labels in Figure 3 and Figure 4 in the main text.  The remaining in Appendix will be fixed before the finalization of the camera-ready version.
>
>
>
> 2.  **The experiments should be conducted on a broader range of datasets.**
>
> - In response to your recommendations, we have included an additional real-world application in Appendix I.  For your convenience, we summarized the experiment here.
>
>
>
> - To illustrate the usage of FISC, we applied our method to MXenes, an early transition metal carbides with two-dimensional (2D) structures exhibiting metallic conductivity and hydrophilicity. These materials are denoted by $M_{n+1}X_{n}T_{x}$, with $M$ representing an early transition metal such as Sc or Ti, $X$ signifying either Carbon or Nitrogen and functional groups $T$ (e.g., O, F, OH) for surface termination. By intercalating different ions and molecules like Li$^{+}$  and K$^{+}$,  MXenes' electrochemical attributes such as voltage, induced charge and capacity can be modulated, serving as a potential battery material.
>
>
>
> - We used a dataset that contains MXenes intercalated with Li$^+$, Na$^+$, K$^+$, and Mg$^{2+}$ ions with properties, voltage, induced charge and capacity. We represented the dataset by their categories, for instance, category Z encoded by 0–3 to describe Li, Na, K, and Mg, respectively, aiming to discover the interactions between different categories. For each property, we trained an MLP  to serve as the reference model and applied our framework to visualize FISC.  The results reveal that FIS exhibits a broad range when category Z is involved in interactions during the prediction of induced charge. This aligns with our common knowledge, given that ions, which form the category Z, are the cause of induced charge, shown in Appendix I. These insightful results offer a more comprehensive understanding of feature interactions within the Rashomon set and potentially guide researchers in making informed decisions for future research.
>
>
>
> 3. **Is your work the first one to propose to use the loss change to measure the strength of feature interactions?**
>
>
>
> - That is a great question. We conducted an investigation and did not find an exact equivalent definition in the existing literature. There are several works with similar ideas. For example, H-statistics [2] measure the strength of pairwise interactions based on the concept of Partial Dependence (PD) and it shows the marginal effect by averaging model outputs, rather than explicit loss like our work. The ANOVA [1] test examines the corresponding p-value for each pair of features as an interaction measurement, which isn't a precise match for our approach. Shapley interaction value and Shapley Taylor Interaction Index [3] are based on the game theory and measure the change in the function value by the addition of features. To our best knowledge, there is no identical definition in the literature, and we can claim that our method is the first one to define the feature interaction range in the Rashomon set.
>
>
>
> 4. **From my perspective, feature interactions typically don't have a definitive "ground truth." Therefore, how can you prove the superiority of your Feature Interaction Score (FIS) compared to other baselines without directly comparing it to a ground truth?**
>
>
>
> - Good point! We appreciate your question and agree that feature interactions typically don't have a definitive "ground truth".
>
> - **(FIS vs FIS)** In certain scenarios, we can explicitly interact features in a function so that we have a "ground truth", such as functions in Synthetic Validation, Section 4.1. These synthetic cases serve as valuable benchmarks for evaluating the accuracy of feature interaction detection methods and facilitate comparisons among different approaches.
>
> - **(FIS vs FISC)** In more general cases, especially in real-world applications that lack a readily available "ground truth" for feature interactions, current methods primarily focus on FIS derived from a single model. As previously noted, the absence of ground truth values in such applications increases the risks associated with relying on a single model. Therefore, we recommend exploring a model class that provides a FISC and their distributions. This approach, in contrast to conventional methods, offers richer insights and a more comprehensive understanding.
>
> - **(FISC vs FISC)** When comparing FISC derived from different searching methods, the analysis typically involves examining FISC statistics such as std, mean, min and max. How to evaluate these comparisons is another interesting direction for further research.
>
>
> **Continuing...**

---

> > ### Author Response · Authors · 2023-11-15
> >
> > 5. **What is MCR in Table 3?**
> >
> > - Thank you for pointing that out. Model class reliance (MCR) refers to the range of feature importance (main effect in our definition) values in the Rashomon set. We followed the name convention, as it aligns with the common practice in Rashomon set-based feature importance exploration. We have added the explanation of MCR in section 4.2.1.
> >
> >
> >
> > 6.  **What is the range of mask? Mask values are used to be 0-1. However, in your algorithm, they seem to be?**
> >
> >
> >
> > - In our definition, the range of mask depends on the selection of epsilon and applications. The mask is not limited to [0,1], rather, it can be assigned any real numbers, provided that the resulting model remains in the Rashomon set. To mitigate the potential confusion for readers, we have added additional explanations in Section 2.1.
> >
> > ---
> >
> > We hope that the above explanations have addressed your concerns. If you have further questions or comments, please let us know.
> >
> >
> >
> > References:
> >
> > [1] Fisher, R. A. (1970). Statistical methods for research workers. In _Breakthroughs in statistics: Methodology and distribution_ (pp. 66-70). New York, NY: Springer New York.
> >
> > [2] Friedman, J. H., & Popescu, B. E. (2008). Predictive learning via rule ensembles.
> >
> > [3] Sundararajan, M., Dhamdhere, K., & Agarwal, A. (2020, November). The shapley taylor interaction index. In _International conference on machine learning_ (pp. 9259-9268). PMLR.

---

> ### Author Response · Authors · 2023-11-20
>
> Thank you for your replies and insightful questions. Here are our responses:
>
> ---
>
> 1.  **Why  $m_+=m_+×(1+lr)$  will result in an upper bound but  $m_−=m_−×(1−lr)$  will result in a lower bound? The upper bound is  max$FIS$?**
>
> - That is a great point. The expressions $m_+=m_+×(1+lr)$ and  $m_−=m_−×(1−lr)$ are used to adjust the values of $m_+$​ and $m_-$​​ respectively. When we say that $m_+=m_+×(1+lr)$ results in an upper bound, we mean that it increases the value of $m$, pushing it towards the upper limit of its possible range. Similarly, $m_−=m_−×(1−lr)$ decreases the value of $m_-$. The terms “upper bound” and “lower bound” in this algorithm do not refer to the maximum and minimum values of the FIS. Instead, they refer to the bounds for the mask values​. The relationship between m, the loss function, and FIS is complex, and the maximum and minimum FIS do not necessarily correspond to the upper and lower bounds of m.
> - To clarify this point and avoid potential confusion, we have added further explanations in the algorithm.
>
>
> 2.  **All dimensional of the mask  m  are applied with the same learning rate  (1−learningrate)  or  (1+learningrate). Does this mean all dimensions of  $X$  are isotropic?**
> - No, it does not mean that all dimensions of X are isotropic. The dimensions of $X$ are determined by the given dataset, and we do not make any assumptions about isotropy in the dataset. The learning rate is initially set to be the same for all dimensions, but we search each dimension **respectively** and it can lead $m$ diverge during the search process. This divergence occurs because the importance of different features can vary, leading to different upper bound and lower bound of $m$ for each feature. Therefore, even though the it starts the same for all dimensions, it quickly becomes specific to each dimension during the search process. The name of learning rate might be the reason that causes the confusion and we will add further explanations to enhance the clarity.
>
> 3.  **In the author's response, the mask  m  can be any real value. In algorithm 1,  m+  can be a positive large value,  m−  can be a positive small value closer to  0. Why they can not be negative?**
> - You’re absolutely correct, and we greatly appreciate your keen observation! The mask $m$ can indeed take any real value, and there was a typo in Algorithm 1. The correct expressions should be $m_−=m_−-lr$ and $m_+=m_+ + lr$. This allows $m_-$​ and $m_+$​ to decrease (below zero) and increase respectively, covering the range of possible real numbers. We apologize for the confusion and appreciate your understanding.
>
>  We hope this addresses your question. If you have any further queries, please let us know. We appreciate your feedback.

---

### Official Review · Reviewer_gxVk · 2023-11-02

**Soundness:** 3 good
**Presentation:** 3 good
**Contribution:** 2 fair
**Rating:** 6
**Confidence:** 3

**Summary:**

This paper presents to explain the feature interactions in a model class. Two visualization tools are developed to analyze the feature interactions.

**Strengths:**

1. Proposed to look for multiple feature interaction sets based on shapley values.
2. Two visualization methods are proposed for analyzing and visualizing the FIS.

**Weaknesses:**

1. I  am not fully convinced by the motivation of this paper. i.e., why do we need to explain feature interactions in a model class?
2. The novelty of Shapley value calculation is not well presented. The focus is totally on the Rashomon set side.

**Questions:**

N.A.

**Details Of Ethics Concerns:**

N.A.

---

> ### Author Response · Authors · 2023-11-14
>
> We appreciate the time and effort you dedicated to reviewing our paper. Your feedback is valuable, and we respect your perspective. However, we would like to address some potential misunderstandings regarding the focus of our paper, which could lead to an underestimation of the significance of our work.
>
> ---
>
>  1. **I am not fully convinced by the motivation of this paper. i.e., why do we need to explain feature interactions in a model class?**
> - Interactions play an important role in model interpretation. Present methods often calculate interactions based on a single well-trained model, yet our demonstration reveals that a well-trained predictive model may not accurately preserve the true feature interactions in Limitations of previous work, Section 1. This discrepancy implies that existing methods could potentially yield inaccurate outcomes. Consequently, our aim is to explore feature interactions within a model class, which greatly increases the chance to uncover the "true" feature interactions. Moreover, with an increasing number of researchers supporting the exploration of a set of equally good models, the feature importance within a model class has gained prominence [1,2,3,4]. Notably, there is a gap of previous work specifically addressing feature interactions in a model class, making it a valuable avenue for exploration.
> - Given the critical role of precise feature interpretation in various scientific research endeavors, the imperative to sidestep inaccurate explanations is paramount to mitigate potential severe consequences. For example,
> 	-  Drug–target interaction (DTI) is vital in drug discovery [5,6]. Predicting the interactions between drugs and targets becomes crucial in the drug discovery task. Different features, such as Enhanced Amino Acid Composition (EAAC) and Position-Specific Scoring Matrix (PSSM), extracted from protein sequences are employed for DTI prediction. These features are input into models for learning and predicting DTI. As the interaction prediction from a single model might be inaccurate, the drug discovery process would lead to misleading outcomes.
> 	- To show the interaction variation in real-world application, we have included another example in Appendix using MXenes, an early transition metal carbide known for its metallic conductivity and hydrophilicity. The distribution of FIS offers valuable insights for researchers, and it can guide researchers in making informed decisions, potentially saving valuable resources that might otherwise be expended on redundant experiments.
>
> 2. **The novelty of Shapley value calculation is not well presented. The focus is totally on the Rashomon set side.**
>
> - We appreciate the acknowledgement of the Shapley value's value, and we extend our gratitude to the reviewer for recognizing our focus on the Rashomon set. Our paper is dedicated to exploring the feature interaction scores within the Rashomon set, and the feature interactions might be defined differently in future research. Importantly, we first propose and define the problem that a well-trained predictive model might not accurately capture the true feature interactions. To prevent that, we recommend exploring feature interaction strengths in a model class of approximately equally accurate predictive models, making the Rashomon set our focus, rather than Shapley value.
>
> ---
>
> We hope that the above explanations have provided a clearer understanding of our motivation and focus, and we are looking forward to hearing back from you.
>
> References:
>
> [1] Fisher, A., Rudin, C., & Dominici, F. (2019). All Models are Wrong, but Many are Useful: Learning a Variable's Importance by Studying an Entire Class of Prediction Models Simultaneously. J. Mach. Learn. Res., 20(177), 1-81.
>
> [2] Rudin, C. (2019). Stop explaining black box machine learning models for high stakes decisions and use interpretable models instead. Nature machine intelligence, 1(5), 206-215.
>
> [3] Zhong, C., Chen, Z., Seltzer, M., & Rudin, C. (2023). Exploring and interacting with the set of good sparse generalized additive models. arXiv e-prints, arXiv-2303.
>
> [4] Xin, R., Zhong, C., Chen, Z., Takagi, T., Seltzer, M., & Rudin, C. (2022). Exploring the whole Rashomon set of sparse decision trees. Advances in Neural Information Processing Systems, 35, 14071-14084.
>
> [5] Sachdev, K., & Gupta, M. K. (2019). A comprehensive review of feature based methods for drug target interaction prediction. Journal of biomedical informatics, 93, 103159.
>
> [6] Abbasi Mesrabadi, H., Faez, K., & Pirgazi, J. (2023). Drug–target interaction prediction based on protein features, using wrapper feature selection. Scientific Reports, 13(1), 3594.

---

> ### Comment · Reviewer_gxVk · 2023-11-23
>
> I agree with the second point, and I will revise the score. Sorry that I am not convinced by this problem setting which looks for a set of explanations. So based on the current version, I cannot recommend acceptance.

---

### Official Review · Reviewer_CnMm · 2023-11-05

**Soundness:** 3 good
**Presentation:** 4 excellent
**Contribution:** 4 excellent
**Rating:** 6
**Confidence:** 4

**Summary:**

The paper proposes feature interaction scores (FIS), where it characterizes feature interactions based on a model class (which achieve similar performance for a task "Rashomon Set") instead of a single model only. Complementary to this, the paper introduces an algorithm that can be used to compute FIS based on Rashomon sets, and a "Halo" plot to visualize the said feature interactions.

Overall, the paper is well written and makes a significant contribution. I have a list of clarifying questions/comments. I would be able to make better analysis of the results based on the discussion. I look forward to author's responses.

**Strengths:**

1. The paper is very well-motivated
2. The proposed method is creative
2. The paper is very clear and easy to read (a few parts need some clarifications; see "Questions"), I congratulate the authors for their clarity of presentation.
3. Makes an important contribution.

**Weaknesses:**

1. The greedy algorithm (one of the main contributions) is unclear and difficult to follow (but can be clarified during discussion)
2. Halo plots are new, and need more discussion/explanation.

**Questions:**

1. In the example where the paper considers roots of a quadractic equation, it is unclear what "the input variables a, b, and c exhibit different feature interactions in these two models" means. Are the "two models" the two roots?  Further explanation is required, it is confusing what the readers are asked to infer.

2. Consider adding a sentence to connect eq (6) to (2), currently it is a sentence fragment.

3. \mathcal{M} is not defined in (7) (its first use).

4. How is eq (6) connected to (7)? It seems that the write-up connecting these two was skipped.

5. Is there a reason to define and call the function in (5) as g(.)? Since the write-up so far uses f(.) for the predictive fuction, can the paper use f(.) instead of g(.)? The rationale is that in (8) the authors define the Rashomon set using g, while (1) uses f. Further, this change means that \mathcal{M} can be defined in (1), and can improve the flow of the paper.

6. What does the paper mean by "then inversely calculate any order of interaction by decomposing..." at the end of 3.2.1.

7. Notation \mathbbm{1}{i=1}^{p} will be more appropriate than $(1)_{i=1}^{p}$ in Algorithm 1. But more importantly, it is unclear how m+ is different from m-. They seem like a vector of size p of ones, i.e. both are the same. If so, then what is the difference between m_i+ and m_i-?

8. Algorithm 1 is somewhat difficult to follow. Since this is the key contribution, can the authors explain the role of various terms such as "learning rate", \phi_s etc. are?
Editing the algorithm to have comments, or updating 3.2.2 to reflect the terms in Alg. 1, can help. For instance, I am still unclear how the computations are taking place over multiple models, and if this is being influenced by "learning rate" somehow.

9. Can the authors explain what they mean by "In theory, the joint effects of features should not exceed the boundary when there is no feature interaction."? Mathematical formulation can help.

10. Which functions is depicted in Fig. 3? We see x_0, x_1, and x_2, but it is unclear what was the original functional form.

11. What does "*" refer to in computational time section?

12. Fig. 4 would benefit from axis labels, and needs to be referred to in the write-up.

13. Halo plots -- the blue curves for a fixed \epsilon account for all \phi_{i,j} (in the 2D case), but what is the x and y axis supposed to denote? Do the negative and positive values on the y axis carry meaning? Why does the x axis not have any axis values?

14. The term "MCR" is undefined.

---

> ### Author Response · Authors · 2023-11-15
>
> We greatly appreciate your support for our work, recognising it as “significant contribution” and “clarity of presentation”. Thank you for all the comments; they have led to a significant improvement of our paper. Concerning the identified weaknesses, we believe that the explanations provided for the proposed questions effectively address your concerns. The following are detailed reply and we have addressed each of these questions in the manuscript (marked in light blue) accordingly
>
> ---
>
>
> 1. **In the example where the paper considers roots of a quadractic equation, it is unclear what "the input variables a, b, and c exhibit different feature interactions in these two models" means. Are the "two models" the two roots? Further explanation is required, it is confusing what the readers are asked to infer.**
>
> - Yes, the “two models” refer to the models for the two roots and they are $f_1(a,b,c) = \frac{-b+\sqrt{b^2-4ac}}{2a}$ and $f_2(a,b,c)=\frac{-b-\sqrt{b^2-4ac}}{2a}$. Both of them are error-free models. Our goal is to demonstrate that these two models induce different feature interaction scores. This is obvious as there is a sign difference in the numerators. More intuitively, consider a special case where $c=0$; the models reduce to $f_1(a,b,c) = 0$ and $f_2(a,b,c)=\frac{-b}{a}$, leading to distinct feature interactions among $a$ and $b$. We have made it clearer in Section 1.
>
>
>
> 2. **Consider adding a sentence to connect eq (6) to (2), currently it is a sentence fragment.**
>
> - Following your suggestion, we have revised sentences in Section 3.1 to connect eq (6) to (2).
>
> 3. **\mathcal{M} is not defined in (7) (its first use).**
>
> - We removed Eq. 7 and referred to Appendix D.0.2.
>
> 4. **How is eq (6) connected to (7)? It seems that the write-up connecting these two was skipped.**
>
> - The middle steps are illustrated in Appendix D.0.2 due to the page limitation. We removed Eq. 7 to improve the flow of the paper and save space.
>
> 5. **Is there a reason to define and call the function in (5) as g(.)? Since the write-up so far uses f(.) for the predictive fuction, can the paper use f(.) instead of g(.)? The rationale is that in (8) the authors define the Rashomon set using g, while (1) uses f. Further, this change means that \mathcal{M} can be defined in (1), and can improve the flow of the paper.**
>
> - Great point. Thank you. We have revised the text accordingly.
>
> 6. **What does the paper mean by "then inversely calculate any order of interaction by decomposing..." at the end of 3.2.1.**
>
> - Given the greedy algorithm, we aim to search the Rashomon set and find main effects for features $(\boldsymbol{m}\_i)\_{i=1}\^{p}$. In order to find interaction effects, e.g., $\{i,j\}$, that requires search masks $m_{i,j}$, which is time-consuming. Instead of searching again, we can utilise the explored mask $m_i$ and $m_j$ and append them together as $(m_i | m_j)$, which is equivalent to $m_{i,j}$ as feature $i$ and feature $j$ are two different features. By doing so, we can inversely calculate any order of interactions.
>
>
>
> 7. **Notation \mathbbm{1}{i=1}^{p} will be more appropriate than $(1)^{p}_{i=1}$ in Algorithm 1. But more importantly, it is unclear how m+ is different from m-. They seem like a vector of size p of ones, i.e. both are the same. If so, then what is the difference between m_i+ and m_i-?**
>
> - We thank the reviewer for the suggestions, and we have incorporated the recommended changes in the format of 1 in Algorithm 1. $m_{i-}$ and $m_{i+}$ are different vectors for the same feature. Initially, $m_{i-}$ and $m_{i+}$ are identical vectors of 1s, but their divergence occurs during the search process. The $m_{i+}$ begins to increase, while the $m_{i-}$ starts decreasing. This distinction arises because our objective is to increase the loss value, and both actions serve this purpose effectively.
>
>
>  **Continuing**...

---

> > ### Author Response · Authors · 2023-11-15
> >
> > 8. **Algorithm 1 is somewhat difficult to follow. Since this is the key contribution, can the authors explain the role of various terms such as "learning rate", \phi_s etc. are? Editing the algorithm to have comments, or updating 3.2.2 to reflect the terms in Alg. 1, can help. For instance, I am still unclear how the computations are taking place over multiple models, and if this is being influenced by "learning rate" somehow.**
> >
> > - Thank you for pointing that out. We have updated section 3.2.2 to reflect the terms in Algorithm 1. As models in the Rashomon can be characterized by masks, we alternatively search masks instead of searching all models. So, each mask corresponds to a model in the Rashomon set. This process is illustrated in Section 3.2.2 and Alg. 1, where we search in the “feature” directions, e.g., $m_i$ for $i$-th feature and $m_j$ for $j$-th feature, to quantify the main effects.
> >
> > - During the search process, we set a learning rate to regulate the quantity of models generated. Higher learning rates lead to a reduced number of models. The searching process continues until the loss condition is not satisfied $\mathcal{M}\_{\boldsymbol{m}\_i} \notin \mathcal{F}$.  This condition is imposed on the loss difference, denoted as $\phi\_{i} = \mathbb{E}[L(\mathcal{M}\_{\boldsymbol{m}\_i}(\boldsymbol{X}), \boldsymbol{y})] - \mathbb{E}[L(f(\boldsymbol{X}), \boldsymbol{y})]$. Any mask $\boldsymbol{m}\_i$ during training meets $\mathcal{M}\_{\boldsymbol{m}\_i} \in \mathcal{F}$ and is one of the target models for feature $\boldsymbol{x}\_i$ in $\mathcal{R}(\epsilon, f\^{*}, \mathcal{F})$.
> >
> >
> >
> >
> >
> > 9. **Can the authors explain what they mean by "In theory, the joint effects of features should not exceed the boundary when there is no feature interaction."? Mathematical formulation can help.**
> >
> > - Good point. Suppose we have $X = [x_i, x_j]$, and we know that $\phi_{i} = L(f \circ m_i(X) - L(f(X)) = L(f(m_i x_i, x_j) - L(f(x_i, x_j))$ and $\phi_{j} = L(f \circ m_j(X) - L(f(X)) = L(f(x_i, m_j x_j) - L(f(x_i, x_j))$, which are used to ensure the Rashomon set loss property in searching process. We can similarly derive $\phi_{i,j} = L(f(m_ix_i, m_j x_j) - L(f(x_i, x_j))$ for their joint effect. Under the assumption that there is no feature interaction between $x_i$ and $x_j$, we can decompose the interaction effect as the sum of individual effects. We denote $g(.)$ as the unknown decomposed function. So we can derive as follows:
> >
> > $$\phi_{i} + \phi_{j} = L(f(m_i x_i, x_j) - L(f(x_i, x_j)) +  L(f(x_i, m_j x_j) - L(f(x_i, x_j)) = g(m_i x_i) + g(x_j) + g(m_j x_j) + g(x_i) = L(f(m_ix_i, m_j x_j) - L(f(x_i, x_j)) = \phi_{i,j}$$
> >
> > - As we have all main effects, we can set $\phi\_{i} + \phi\_{j} \leq \epsilon$ and we can derive $\phi\_{i,j} \leq \epsilon$, which indicates that the joint effect is within the boundary. We have rephrased this sentence in the main text, Section 3.2.3, to avoid confusion.
> >
> > 10. **Which functions is depicted in Fig. 3? We see x_0, x_1, and x_2, but it is unclear what was the original functional form.**
> >
> > - The function depicted in Fig. 3 is $f(x) = \bigwedge (x; \{x\_0^*, x\_1^* \} \cup x'\_2 ) + \bigwedge (x; \{x\_i\^{*}\}\_{i=11}^{30}) +\sum_{j=1}^{40}x_j$ from Table 1 in Section 4.1.  To provide clarity on the context, we have included the function in the caption of Fig. 3.
> >
> > 11. **What does "*" refer to in computational time section?**
> >
> > - The asterisk (*) used here indicates multiplication, and we have revised the sentence to minimize potential confusion.
> >
> > 12. **Fig. 4 would benefit from axis labels, and  needs to be referred to in the write-up.**
> >
> > - Thank you. We have addressed it by incorporating axis labels to enhance clarity and this figure is referred to in the "Going beyond the literature", Section 4.2.1.
> >
> > **Continuing**...

---

> > > ### Author Response · Authors · 2023-11-15
> > >
> > > 13. **Halo plots -- the blue curves for a fixed \epsilon account for all \phi_{i,j} (in the 2D case), but what is the x and y axis supposed to denote? Do the negative and positive values on the y axis carry meaning? Why does the x axis not have any axis values?**
> > >
> > > - This is an excellent question. $x$ and $y$ axis are decomposed $\phi_{i,j}$ into $\phi\_{i}$ and $\phi\_{j}$ directions. Remember that $\phi\_{i,j}(\mathcal{M}\_{\boldsymbol{m}\_{i, j}}, f)$ is a function of $m_i$, $m_j$ and an unknown decomposed function $g(.)$. The red circle can be seen as the expected (without interaction) value of $m_i$, $m_j$ in $g(m_i, m_j) = \phi_{i} + \phi_{j}$, while the blue circle is the empirical value (with interaction) of same $m_i$, $m_j$, but in function $\hat{g}(m_i, m_j) = \phi_{i,j}$.  Here we only use the 2-D space to represent it, as the $g(.)$ and $\hat{g}(.)$ can be calculated from empirical loss.
> > >
> > > - The positive and negative values on the y axis represent the searching direction only. For each $m_i$ and $m_j$, we have two searching directions, denoted as $m_{i+}$, $m_{i-}$, $m_{j+}$, $m_{j-}$, referring to the algorithm, and both actions increase loss values. Thus, we can derive $\phi_{i+} + \phi_{j+} = \phi_{i+} + \phi_{j-} = \phi_{i-} + \phi_{j+} = \phi_{i-} + \phi_{j-}$. We have added values in x axis in figures to clarify the representation.
> > >
> > > 14. **The term "MCR" is undefined.**
> > >
> > > - Thank you for pointing that out. Model class reliance (MCR) refers to the range of feature importance (main effect in our definition) values in the Rashomon set. We followed the name convention, as it aligns with the common practice in Rashomon set-based feature importance exploration. We have added the explanation of MCR in section 4.2.1.
> > >
> > >  ---
> > >
> > > Once again, we are grateful for your comments. We hope that the above explanations have addressed your concerns. If you have further questions or comments, feel free to let us know. We look forward to hearing you back.

---

### Author Response · Authors · 2023-11-15

We would like to thank all reviewers’ time and efforts in reviewing our paper. We appreciate that the reviewers found the paper novel and well-presented. All reviewers' major and minor comments are addressed accordingly in manuscript (marked in light blue), and we are dedicated to resolving any further questions that may arise.

---

### Author Response · Authors · 2023-11-21

Dear reviewers,

---

We would like to thank you for your valuable insights and feedback on our paper so far. We appreciate your time and effort in reviewing our work and providing constructive comments.

We hope our responses have clarified your queries and addressed your concerns. We would like to answer any potential questions and finalize our paper before the discussion deadline **tomorrow**. Please do not hesitate to contact us if you have any questions or suggestions for improvement.

We are grateful for your recognition and continued support.

---

Sincerely,

The Authors

---

### Meta-Review · Area_Chair_9Muo · 2023-12-06

**Metareview:**

In this paper, the authors address the critical role of feature interactions in understanding the behaviors of machine learning models. While recent research has advanced in detecting and quantifying these interactions within single predictive models, the authors argue that interactions derived from a single, well-trained model may not accurately capture true feature dynamics. This is because different high-performing models can demonstrate varied strengths in feature interactions. To overcome this limitation, the authors propose exploring feature interaction strengths across a collection of models with similar accuracy, termed a Rashomon set. They introduce a new metric, the Feature Interaction Score (FIS), specifically designed for use within this Rashomon set. The paper presents a general and practical algorithm for calculating the FIS across this model class. To illustrate the utility and properties of the FIS, the authors employ synthetic data and establish connections with broader statistical concepts. They also introduce innovative visualization tools, including the Halo plot, which visualizes feature interaction variance in high-dimensional space, and the swarm plot, for analyzing FIS within the Rashomon set. Through empirical studies in recidivism prediction and image classification, the authors demonstrate that feature interactions can vary significantly in their importance across models with comparable predictive accuracy. The findings suggest that the proposed FIS offers valuable insights into the nature of feature interactions in machine learning models.

In the initial review and the second review, no reviewer is against the novelty of this paper, which further confirms that this paper is novel and significant in its field. Based on the high novelty this paper contains, I recommend accepting this paper at the current stage. However, the authors are suggested to further strengthen the presentation of this paper. One reviewer actually suggested rejection due to a significant typo that should be definitely avoided in the final revision.

**Justification For Why Not Higher Score:**

The presentation of this paper should be further improved. This is not a perfect paper but is a novel paper.

**Justification For Why Not Lower Score:**

In the initial review and the second review, no reviewer is against the novelty of this paper, which further confirms that this paper is novel and significant in its field. Based on the high novelty this paper contains, I recommend accepting this paper at the current stage.

---

### Decision · Program_Chairs · 2024-01-16

Accept (poster)